



Atmospheric
Chemistry
and Physics

# Vertical variability of the properties of highly aged biomass burning aerosol transported over the southeast Atlantic during CLARIFY-2017

**Huihui Wu[1], Jonathan W. Taylor[1], Kate Szpek[2], Justin M. Langridge[2], Paul I. Williams[1,3], Michael Flynn[1], James D. Allan[1,3], Steven J. Abel[2], Joseph Pitt[1,a], Michael I. Cotterell[4,b], Cathryn Fox[2], Nicholas W. Davies[2,4], Jim Haywood[2,4], and Hugh Coe[1]**

[1]Department of Earth and Environmental Sciences, University of Manchester, Manchester, UK
[2]Met Office, Fitzroy Road, Exeter, EX1 3PB, UK
[3]National Centre for Atmospheric Science, University of Manchester, Manchester, UK
[4]College of Engineering, Mathematics and Physical Sciences, University of Exeter, Exeter, UK
[a]now at: School of Marine and Atmospheric Science, Stony Brook University, Stony Brook, USA
[b]now at: School of Chemistry, University of Bristol, Bristol, BS8 1TS, UK

**Correspondence:** Hugh Coe (hugh.coe@manchester.ac.uk)

**Abstract.** TS1 Seasonal biomass burning (BB) from June to October in central and southern Africa leads to absorbing aerosols being transported over the South Atlantic Ocean every year and contributes significantly to the regional climate forcing. The vertical distribution of submicron aerosols and their properties were characterized over the remote southeast Atlantic, using airborne in situ measurements made during the CLoud-Aerosol-Radiation Interactions and Forcing for Year 2017 (CLARIFY-2017) campaign. BB aerosols emitted from flaming-controlled fires were intensively observed in the region surrounding Ascension Island, in the marine boundary layer (MBL) and free troposphere (FT) up to 5 km. We show that the aerosols had undergone a significant ageing process during > 7 d transit from source, as indicated by the highly oxidized organic aerosol. The highly aged BB aerosols in the far-field CLARIFY region were also especially rich in black carbon (BC), with relatively low single-scattering albedos (SSAs), compared with those from other BB transported regions.

The column-weighted dry SSAs during CLARIFY were observed to be 0.85, 0.84 and 0.83 at 405, 550 and 658 nm respectively. We also found significant vertical variation in the dry SSA, as a function of relative chemical composition and size. The lowest SSA in the column was generally in the low FT layer around 2000 m altitude (averages: 0.82, 0.81 and 0.79 at 405, 550 and 658 nm). This finding is important since it means that BB aerosols across the southeast Atlantic region are more absorbing than currently represented in climate models, implying that the radiative forcing from BB may be more strongly positive than previously thought. Furthermore, in the FT, average SSAs at 405, 550 and 658 nm increased to 0.87, 0.86 and 0.85 with altitude up to 5 km. This was associated with an enhanced inorganic nitrate mass fraction and aerosol size, likely resulting from increased partitioning of ammonium nitrate to the existing particles at higher altitude with lower temperature and higher relative humidity. After entrainment into the boundary layer (BL), aerosols were generally smaller in dry size than in the FT and had a larger fraction of scattering material with resultant higher average dry SSA, mostly due to marine emissions and aerosol removal by drizzle. In the BL, the SSA decreased from the surface to the BL top, with the highest SSA in the column observed near the surface. Our results provide unique observational constraints on aerosol parameterizations used in modelling regional radiation interactions over this important region. We recommend that future work should consider the impact of this vertical variability on climate models.

**Published by Copernicus Publications on behalf of the European Geosciences Union.**

# 1  Introduction

Open biomass burning (BB) is a major source of global trace gases and carbonaceous aerosol particles in the atmosphere. The smoke aerosol emitted from BB is mainly comprised of strongly absorbing black carbon (BC) and fine organic aerosol (OA), whose proportions vary according to vegetation type, oxygen availability and combustion phase (Andreae and Merlet, 2001; Andreae, 2019). As controls continue to reduce aerosol emissions from fossil fuels and a changing climate potentially leads to more fires, the relative impact of BB on climate forcing is expected to increase (Fuzzi et al., 2015).

Seasonal burning of grasslands and agricultural residue occurs between June and October across the central and southern African savanna, contributing about one-third of the global BB emissions (van der Werf et al., 2010). Previous space-based observations showed that smoke aerosols produced by this burning are primarily transported westward for thousands of kilometres over the South Atlantic region by free tropospheric (FT) winds (Edwards et al., 2006; Adebiyi and Zuidema, 2016). These smoke layers typically over-lie vast stretches of marine stratocumulus clouds (Adebiyi et al., 2015), where they can exert a warming effect by absorbing both downwelling solar radiation and that scattered upwards from the low-lying clouds (Samset et al., 2013). This direct radiative effect is sensitive to the smoke's single-scattering albedo (SSA), which is a function of aerosol composition and size and evolves with particle age (Abel et al., 2005). Space-based and in situ field observations also suggested that the smoke layers can be entrained into the marine boundary layer (MBL) during its transport from land over ocean (Painemal et al., 2014; Zuidema et al., 2018; Haslett et al., 2019a). The entrained aerosols in the MBL can affect cloud microphysics by acting as cloud condensation nuclei (CCN), inducing indirect radiative effects over the southeast Atlantic by increasing cloud droplet number and reducing precipitation, thereby increasing cloud coverage and cloud albedo (e.g. Costantino and Bréon, 2013). In addition, BC immersed in cloud droplets absorbs light and may facilitate water evaporation. BC below clouds could enhance the formation of convection by providing additional heating within the sub-cloud layer. Zhang and Zuidema (2019) reported that short-wave absorption within the smoky MBL reduces the sub-cloud relative humidity due to raising the temperature and so reduces daytime low-level cloud cover over the southeast Atlantic, which is opposite to the mechanism of increased aerosol increasing cloud droplet numbers. Furthermore, large eddy model studies have shown that marine stratocumulus clouds over the southeast Atlantic also adjust to the presence of overlying absorbing aerosol layers, depending on their properties and distance with low-cloud deck (e.g. Herbert et al., 2020). The above-cloud shortwave absorption can warm the FT, strengthening the temperature inversion and reducing the entrainment of warm and dry air from the FT

into the MBL, thus influencing MBL humidity, temperature and dynamics. These effects described above, which perturb the temperature structure of the atmosphere and influence the cloud distribution, are collectively termed semi-direct effects.

BB emission in Africa has been shown to be relatively stable on multi-annual timescales (Voulgarakis et al., 2015), implying that transport of African BB aerosol (BBA) across the Atlantic region has likely been a consistent phenomenon over past decades. Although BB transport regions have lower aerosol concentrations than areas closer to the source, the large spatial coverage means that their contribution to the regional/global-average forcing is important. Moreover, the southeast Atlantic has persistent overlying semi-permanent stratocumulus clouds, and therefore aerosol cloud interactions in this specific transport region are strong. Gordon et al. (2018) simulated the radiative effects of smoke aerosols transported from Africa over the southeast Atlantic area near Ascension Island in a regional model, reporting substantial regional direct radiative effects of $+11\,\mathrm{W\,m^{-2}}$, a semi-direct effect of $-30.5\,\mathrm{W\,m^{-2}}$ and an indirect effect of $-10.1\,\mathrm{W\,m^{-2}}$. This implies an overall cooling effect and highlights the important climate effect of transported BBA over the southeast Atlantic region. The extent to which smoke layers over the Atlantic Ocean subside and entrain into the MBL varies between different models (Peers et al., 2016; Das et al., 2017). Some modelled BB smoke layers quickly descend to lower levels just off the western coast of the continent, whereas space-based observations suggest that smoke layers continue their horizontal transport at elevated levels above the MBL for thousands of kilometres (Das et al., 2017). This is crucial because the simulated aerosol effects are dependent on the vertical distribution of aerosol (especially with respect to clouds) and whether the absorbing aerosols is present within, below or above the cloud (Samset et al., 2013). The uncertainty in simulated aerosol vertical distribution would cause a significant diversity in modelled climate forcing over the region. A recent study demonstrated that models generally underestimate the smoke base height over the southeast Atlantic and thus lead to an overestimation of aerosol loading in the MBL (Shinozuka et al., 2020). Uncertainty in SSA is also one of the largest sources of uncertainty in estimating the aerosol direct effects (McComiskey et al., 2008; Shinozuka et al., 2020). To improve simulations of aerosol radiative effects, it is vital to constrain models using observational studies.

Satellite-based observations have been employed in this region, but satellite retrievals often detect the bottom of the aerosol layer too high and thereby overestimate the above-cloud aerosol height (e.g. Rajapakshe et al., 2017). The ability of satellites to quantify BBA amount and its microphysical and optical properties in the marine BL is also limited, since the presence of intervening cloud layers brings significant challenges to retrievals of aerosol properties. Due to the persistent stratocumulus cloud deck over the South Atlantic,

most of the region is affected by clouds, and so MBL properties are hard to obtain from satellites. Furthermore, satellite retrievals provide column-integrated aerosol properties and fail to provide information on the large vertical variabilities in aerosol properties. The LASIC (Layered Atlantic Smoke Interactions with Clouds) field campaign was conducted on Ascension Island in the southeast Atlantic, delivering in situ ground-based aerosol measurements (Zuidema et al., 2018) and column information retrieved from surface-based remote sensing. These measurements were limited to single-point or column observations but provide a long and continuous time series. The vertically resolved retrievals obtained during the LASIC campaign using a co-located micropulse lidar also have retrieval limitations (Delgadillo et al., 2018).

Previous key aircraft measurements focusing on the southern African BB include the SAFARI 2000 (the Southern African Regional Science Initiative) campaign in September 2000 (Haywood et al., 2003a, b). Fresh BB smoke in SAFARI 2000 was observed on a single flight directly over a terrestrial large fire (on 13 September 2000), aged smoke was observed from flights over the continent or near the Namibian coast, and a single profile of BBA was analysed close to Ascension Island. More recently, the NASA ORACLES (ObseRvations of Aerosols above CLouds and their intEractionS) campaigns in September 2016, August to September 2017, and October 2018 extended measurements over the South Atlantic, mostly sampling westward of the SAFARI region and eastward of 0° E (Zuidema et al., 2016; Pistone et al., 2019). The AEROCLO-sA (Aerosols, Radiation and Clouds in southern Africa) campaign in August to September 2017 also focused on BBA just before crossing the Namibian coast (Formenti et al., 2019). The DACCIWA (Dynamics-Aerosol-Chemistry-Cloud Interactions in West Africa) campaign in June to July 2016 reported aged BBAs that were transported from southern Africa to both the FT and MBL near the southern coastal region of West Africa (Haslett et al., 2019a, b). Although these aircraft measurements covered African BBA with different ages, they did not provide a broad-scale picture of long-range transported BBA over the remote southeast Atlantic. Observations of the vertical distribution of transported BBA over the remote southeast Atlantic are therefore essential to provide better constraints on future climate model studies in this region.

This study uses data from the CLARIFY-2017 (CLoud-Aerosol-Radiation Interactions and Forcing for Year 2017) aircraft campaign, which was conducted in August–September 2017, based from Ascension Island in the southeast Atlantic Ocean. The spatial distribution of MODIS-detected fires for August 2017 is shown in Fig. 1, with the average wind fields at 925 and 700 hPa corresponding to levels in the BL and FT separately. The burning mostly occurred in central and southern Africa (0–20° S) during the campaign period. Large quantities of BBA generally occur in a deep, turbulent, surface-heating-driven layer extending to between 3 and 4.5 km (Labonne et al., 2007). The smoke is then ad-

vected westward over the Atlantic Ocean by the southerly branch of the African easterly jet, as seen in the wind field at around 700 hPa in Fig. 1. The typical atmospheric BL flow, as indicated by the 925 hPa wind field in Fig. 1, follows the climatological wind pattern of south-easterlies, advecting clean Southern Hemisphere air around the southern Atlantic subtropical anticyclone. When smoke is transported from Africa over the South Atlantic it encounters the BL that has deepened further offshore and north of 5° S (Das et al., 2017). Subsiding smoke layers can be entrained into the BL and mix with clean air masses that are transported from the southeast to northwest over the Atlantic Ocean. Typically, smoke plumes have undergone at least 7 d transport since emission before arriving in the BL around Ascension Island (Gordon et al., 2018; Zuidema et al., 2018). Haywood et al. (2020) conducted back trajectories with particles released from Ascension Island at different altitudes from the MBL to FT, and they reported that air masses sampled in the CLARIFY operating area were of African BB origin and also indicated that the aerosol age was likely in a range of 4 to 10 d. The CLARIFY aircraft campaign provides the opportunity to observe vertical structures of African BBA transported to the far-field region over the southeast Atlantic.

This paper presents a synthesis of in situ airborne measurements, including the vertical distribution of submicron aerosols; their chemical, physical, and optical properties; and mixing state using the CLARIFY measurements. We use this analysis to investigate the main factors influencing BBA properties over the southeast Atlantic after long-range transport.

## 2 Methodology

### 2.1 Airborne measurements

The measurements described here were made using the UK Facility for Airborne Atmospheric Measurements (FAAM) BAe-146 Atmospheric Research Aircraft (ARA), which was based out of Ascension Island (7.93° S, 14.42° W) in the southeast Atlantic, as part of the CLARIFY project. 28 scientific flights (designated flight labels from C028 to C055) took place between 16 August and 7 September 2017. A series of straight and level runs (SLRs) and vertical profiles were performed during each flight. The flight tracks during the campaign are shown in Fig. 1a. Transit flights, C040-41, which took place on 26 August are not included since the aircraft was predominantly in clean air at high altitude. A summary of the flights and scientific deployment is provided by Haywood et al. (2020), while relevant instruments used in this study are discussed in more detail here.

The BAe-146 facility can provide aircraft position information and conducts routine measurements of standard atmospheric variables, such as temperature, pressure and winds. Humidity is measured by a CR-2 chilled mirror hy-

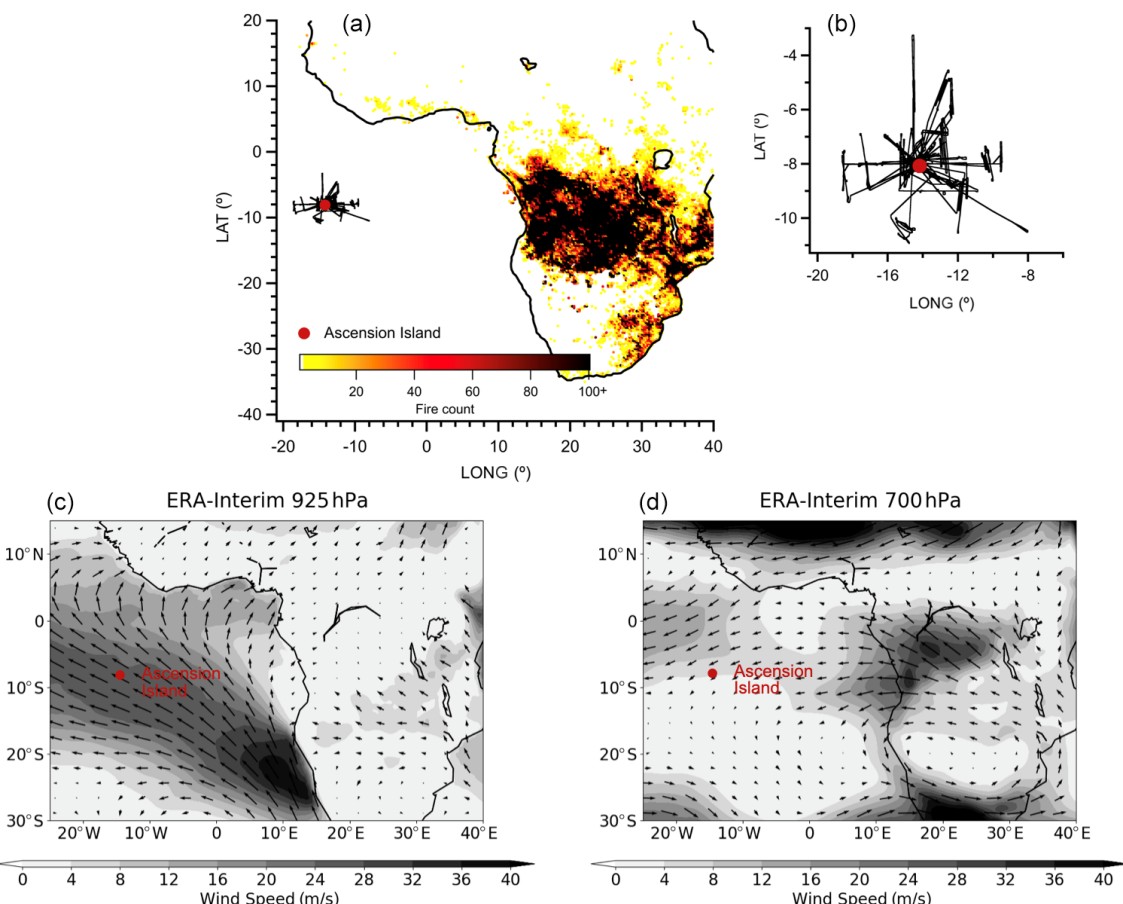

Figure 1. (a, b) The integrated spatial distribution of MODIS-detected fire counts in August 2017, coupled with flight tracks (without transit flights) during the CLARIFY-2017 aircraft campaign (16 August–7 September). (c, d) Average wind speed and direction at 925 hPa (c) and 700 hPa (d) in ERA-Interim reanalysis for August 2017. The wind speed is shown in greyscale bars.

grometer. The inboard instruments used in this study drew their sample via standard BAe-146 Rosemount inlets, which have sampling efficiencies close to unity for submicron particles (Trembath et al., 2012).

The chemical composition of non-refractory submicron aerosols was measured by a compact time-of-flight aerosol mass spectrometer (C-ToF-AMS, Aerodyne Research Inc, Billerica, MA, USA) (Drewnick et al., 2005), which provides chemical characterization across a range of ion mass-to-charge ($m/z$) ratios from 10 to 500. The detailed operation of the AMS, including calibration and correction factors, during aircraft deployment has been described previously (Morgan et al., 2009). The AMS was calibrated using monodisperse ammonium nitrate and ammonium sulfate particles. The AMS data were processed using the standard SQUIRREL (SeQUential Igor data RetRiEvaL, v.1.60N) ToF-AMS software package. A time- and composition-dependent collection efficiency (CE) was applied to the data based on the algorithm by Middlebrook et al. (2012). The uncertainties of mass concentrations from aircraft AMS are estimated in Bahreini et al. (2009). In this study, the mass concentrations

of organic aerosol (OA), sulfate, nitrate and ammonium are determined, and markers ($m/z$ 60, $m/z$ 43 and $m/z$ 44) were used to provide information on the composition of the OA fraction. Proportional contributions of OA fragment markers, $f$60, $f$43 and $f$44, were calculated as the ratios of $m/z$ 60, $m/z$ 43 and $m/z$ 44 to the total OA mass concentration respectively (Ng et al., 2011; Cubison et al., 2011; Ortega et al., 2013). The AMS suffered from a blockage of the inlet during some periods, and data collected from six flights (C042–C044, C052, C054 and C055) are not available.

The refractory black carbon (rBC; hereafter referred to as BC) was characterized using a single particle soot photometer (SP2, Droplet Measurement Technologies, Boulder, CO, USA). The instrument setup, operation and data interpretation procedures can be found elsewhere (McMeeking et al., 2010; Liu et al., 2010). The SP2 can measure BC-containing particles equivalent to a spherical diameter of 70–850 nm (Liu et al., 2010; Adachi et al., 2016). The SP2 incandescence signal is proportional to the mass of refractory BC present in the particle, regardless of mixing state. The SP2 incandescence signal was calibrated us-

ing Aquadag black carbon particle standards (Aqueous Deflocculated Acheson Graphite, manufactured by Acheson Inc., USA), including the correction (0.75) recommended by Laborde et al. (2012a). The overall uncertainty of the BC mass concentration calibration is $\pm 20\%$ (Laborde et al., 2012a, b).

Aerosol number size distribution was measured via two wing-mounted passive cavity aerosol spectrometer probes (PCASP) and an on-board scanning mobility particle sizer (SMPS). The PCASP uses the intensity of scattered light to measure the size of a particle at 1 Hz, over a nominal diameter range of 0.1–3 µm across 30 channels. Particle size is determined via calibrations using di-ethyl-hexyl-sebacate (DEHS) and a polystyrene latex sphere (PSL) with known size and refractive index (Rosenberg et al., 2012). Mie scattering theory was used to determine the bin sizes by assuming particles are spherical, with a refractive index of $1.54 - 0.027i$. The refractive index was obtained by the methods reported by Peers et al. (2019), where the aerosol model best represents the PCASP measurement. The SMPS sampled from the same inlet as the AMS measured distributions of particle mobility diameter divided into 26 or 31 logarithmically spaced bins in the range of 20–350 nm. A low-pressure water-based condensation particle counter (WCPC model 3786-LP) was connected to a TSI 3081 differential mobility analyser (DMA). The SMPS data were inverted using the scheme developed by Zhou (2001), based on a $\sim 1$ min averaging time only during straight and level runs when AMS and SP2 concentrations generally varied less than 20 %. The combination of SMPS and PCASP measurements was used to determine size distributions from 20 nm to 3 µm, providing information on the sub-Aitken and accumulation mode aerosol.

A comparison of the estimated volumes from the AMS and SP2 with the PCASP was conducted, following the method in Morgan et al. (2010). The total mass concentrations measured from the AMS and SP2 were converted to total volume concentrations, using densities of $1.27\,\mathrm{g\,cm^{-3}}$ for organics, $1.77\,\mathrm{g\,cm^{-3}}$ for inorganics and $1.8\,\mathrm{g\,cm^{-3}}$ for BC (Morgan et al., 2010; Liu et al., 2010). The submicron volume concentrations from PCASP were estimated using bins with diameter below 1 µm, assuming particles are spherical. The estimated volumes from the AMS and SP2 were 83 % and 77 % of the estimated PCASP volumes in the BL and FT respectively. These discrepancies are considered tolerable given the 30 %–50 % uncertainty in PCASP volume estimates (e.g. Moore et al., 2004) and the uncertainty in densities required to convert the AMS mass to volume.

The aerosol dry extinction and absorption were measured with the EXSCALABAR instrument (EXtinction, SCattering and Absorption of Light for AirBorne Aerosol Research) which has been developed by the Met Office for use on the ARA (Davies et al., 2018, 2019). It consists of an array of spectrometers making use of photoacoustic spectroscopy (PAS) and cavity ring-down spectroscopy (CRDS)

techniques. The dry (RH < 10 %) aerosol absorption coefficient is measured at wavelengths 405, 514 and 658 nm, and the dry extinction coefficient is measured at wavelengths 405 and 658 nm. An impactor ensures any aerosol with aerodynamic diameter greater than 1.3 µm is removed from the sample. The instrument, including the PAS calibration method, is described in detail by Davies et al. (2018) and Cotterell et al. (2019). The relative contributions of scattering and absorption are given by the single-scattering albedo (SSA), which is calculated as

$$\mathrm{SSA}(\lambda) = 1 - \frac{B_{\mathrm{Abs}}(\lambda)}{B_{\mathrm{Ext}}(\lambda)}, \tag{1}$$

in which $B_{\mathrm{Abs}}$ is the light-absorption coefficient measured by PAS, $B_{\mathrm{Ext}}$ is the light-extinction coefficient measured by CRDS and $\lambda$ is the wavelength. The uncertainty in the SSA calculations is related to the corresponding uncertainties in the extinction and absorption coefficient measurements. The mean SSA uncertainties are determined to be 0.013 and 0.018 at the wavelengths of 405 and 658 nm respectively when only considering systematic errors (Peers et al., 2019).

Carbon monoxide (CO) was measured by a vacuum ultra-violet florescence spectroscopy (AL5002, Aerolaser GmbH, Germany), with an accuracy of $\pm 3\%$ and a precision of 1 ppbv (Gerbig et al., 1999). Calibration was performed using in-flight measurements of a single gas standard and the background signal at zero CO mole fraction. Carbon dioxide ($CO_2$) was measured using a Fast Greenhouse Gas Analyzer (FGGA; Los Gatos Research, USA). The instrument setup, operation and performance on the ARA has been described for several previous aircraft campaigns (O'Shea et al., 2013). The FGGA was calibrated hourly in flight, using a calibration gas standard traceable to the WMO-X2007 scale for $CO_2$. Liquid water content (LWC) was calculated from 1 Hz measurements by the Cloud Droplet Probe (CDP), with the operation and calibration of the CDP described in Lance (2012). An LWC value of $0.01\,\mathrm{g\,m^{-3}}$ was used to define the low threshold for the presence of cloud.

## 2.2 Data analysis and classification

All measurements reported here were corrected to standard temperature and pressure (STP, 273.15 K and 1013.25 hPa), and in-cloud data were removed. SP2, PAS, CRDS, CO, $CO_2$ and PCASP data were recorded at 1 Hz and were averaged onto the AMS time base, which recorded data about every 8–9 s. SSA was calculated from the averaged PAS and CRDS data. Submicron aerosol ($PM_1$) number concentrations from the PCASP were calculated using bins with diameter below 1 µm. SMPS and PCASP size distributions were averaged over each SLR. Flights with the AMS sampling problem mentioned above (C042–C044, C052, C054 and C055), sampling mainly in-cloud (C052–C054) or the transits (C040–C041), are not considered in the following analysis. Flights used in this study are listed in Table S1 in the Supplement.

Over the southeast Atlantic, there is typically a strong thermodynamic inversion at the top of the BL (e.g. Lock et al., 2000). The profiles of temperature and specific humidity were derived from all the flights used in this study (C028–C039, C045–C051), as seen in Fig. S1. The lack of variability shown by the bars demonstrates the ubiquitous nature of this inversion. Here, we define the BL top to be coincident with the base of the temperature inversion, typically at an altitude around 1400–2000 m. The inversion layer sits immediately above the BL and is characterized by a sharp increase in temperature and coincident steep decrease in specific humidity, typically in a thickness range of 100–400 m. Above the inversion layer, the air is dry (specific humidity $< 0.002\,\mathrm{g\,kg^{-1}}$ TS2 compared to $> 0.01\,\mathrm{g\,kg^{-1}}$ TS3 in the BL) and is regarded as being in the FT. Using these thermodynamic criteria, we divided the data from each flight into three parts: the BL, the inversion layer and the FT. The inversion layer data are in the transition between the BL below and FT aloft, and since their characteristics cannot easily be classified, these data are not used in further analysis. In this study, the air masses perturbed by BB pollutants were identified when $BC > 0.1\,\mathrm{\mu g\,m^{-3}}$ to prevent the noise at low aerosol concentrations affecting our analysis. Clean BL air masses were selected when $CO < 66$ ppbv ($53\,\mathrm{\mu g\,m^{-3}}$), which corresponds to the lowest 5th percentile of all CO data collected in the BL.

## 3 Results

Figure 2 shows the average vertical distribution of submicron ($PM_1$, $\mathrm{\mu g\,m^{-3}}$) aerosol mass concentration for each flight. $PM_1$ mass concentration was calculated from the AMS non-refractory submicron species and BC mass from the SP2. During the month-long campaign, there was significant variability in measured aerosol loadings at different layers. Three distinct types of aerosol vertical structures were observed, and consequently we divided the campaign into three periods. From 16 to 19 August (period 1, C028–C032), BBA was concentrated in the BL. During period 2 (from 22 to 25 August, C033–C039), the FT was BB-polluted, and the BL was mostly clean. During period 3 (from 29 August to 5 September, C045–C051), the BB pollution was observed throughout the BL and FT. The following aerosol characterizations (chemical, physical and optical properties) were divided into these periods and different vertical layers (the FT and the BL).

### 3.1 Aerosol chemical properties

In this section, we consider the chemical composition of observed $PM_1$ during CLARIFY and percentage contribution of different components to the total mass. We also investigate the vertical variability of the fractional chemical composition. The OA markers and elemental analysis are used to indicate the properties and ageing status of observed organics. The enhancement ratios of BC and OA were also calculated to obtain some information on the emission conditions at source and the removal during transport (Yokelson et al., 2013).

### 3.1.1 Submicron aerosol compositions

Average composition ratios of BL and FT aerosols for each period are summarized in Table 1, with campaign-average pie charts shown in Fig. 3. Detailed vertical distributions of concentrations of different chemical components in each flight are shown in Fig. S2 in the Supplement. In the BB-polluted FT (periods 2 and 3), the relative chemical composition was similar between flights and periods. The composition fractions (average ± standard deviation) were $(61 \pm 5)\,\%$, $(13 \pm 3)\,\%$, $(11 \pm 4)\,\%$, $(8 \pm 3)\,\%$ and $(7 \pm 2)\,\%$ for OA, BC, sulfate, nitrate and ammonium respectively. In the BB-polluted BL (periods 1 and 3), chemical composition ratios showed temporal variations. The BL in period 1 had $\sim 10\,\%$ higher average sulfate mass fraction and $\sim 6\,\%$ lower BC mass fraction than in period 3. The relative chemical compositions in the BB-polluted BL and FT also showed differences. Sulfate average mass fractions in the BL were $(30 \pm 4)\,\%$ in period 1 and $(21 \pm 5)\,\%$ in period 3, which were 2–3 times larger than those in the FT ($(11 \pm 4)\,\%$). BL ammonium mass fraction was also slightly higher than in the FT. The linear fitted $NH_{4mea}^{+}/NH_{4neu}^{+}$ ratios in the BL were $(0.86 \pm 0.01)$ and $(0.99 \pm 0.02)$ for period 1 and 3 respectively, indicating the possible presence of acidic aerosol during the first period ($NH_{4mea}^{+}$ is the measured ammonium concentration from the AMS, and $NH_{4neu}^{+}$ is the calculated ammonium concentration if all acids in the aerosol were neutralized) (Zhang et al., 2007). When sulfate is not fully neutralized, nitrate aerosol formation is suppressed due to the absence of excess of ammonia. OA and BC accounted for smaller fractions of $PM_1$ in the BL than in the FT.

In the clean BL air masses encountered during period 2, which were representative of a background marine environment, the submicron particle mass ($0.23 \pm 0.18\,\mathrm{\mu g\,m^{-3}}$) was dominated by sulfate ($0.14 \pm 0.10\,\mathrm{\mu g\,m^{-3}}$), with small amounts of OA ($0.06 \pm 0.07\,\mathrm{\mu g\,m^{-3}}$) and negligible other components (see chemical fractions in Table 1). Sulfate mass loadings were significantly enhanced ($1.9 \pm 0.5\,\mathrm{\mu g\,m^{-3}}$ in period 1 and $0.7 \pm 0.2\,\mathrm{\mu g\,m^{-3}}$ in period 3), and other aerosol species were present when BB smoke was transported into the BL. Contribution from the marine sulfate background may explain the higher sulfate fraction reported for the BL BBA than the FT. During the DACCIWA project, sampling near the southern coastal region of West African, aircraft observations showed that the sulfate mass fraction was also enhanced in BL BBA compared with the FT BB layer (see Table 1), after long-range transport of southern African BB smoke (Haslett et al., 2019b).

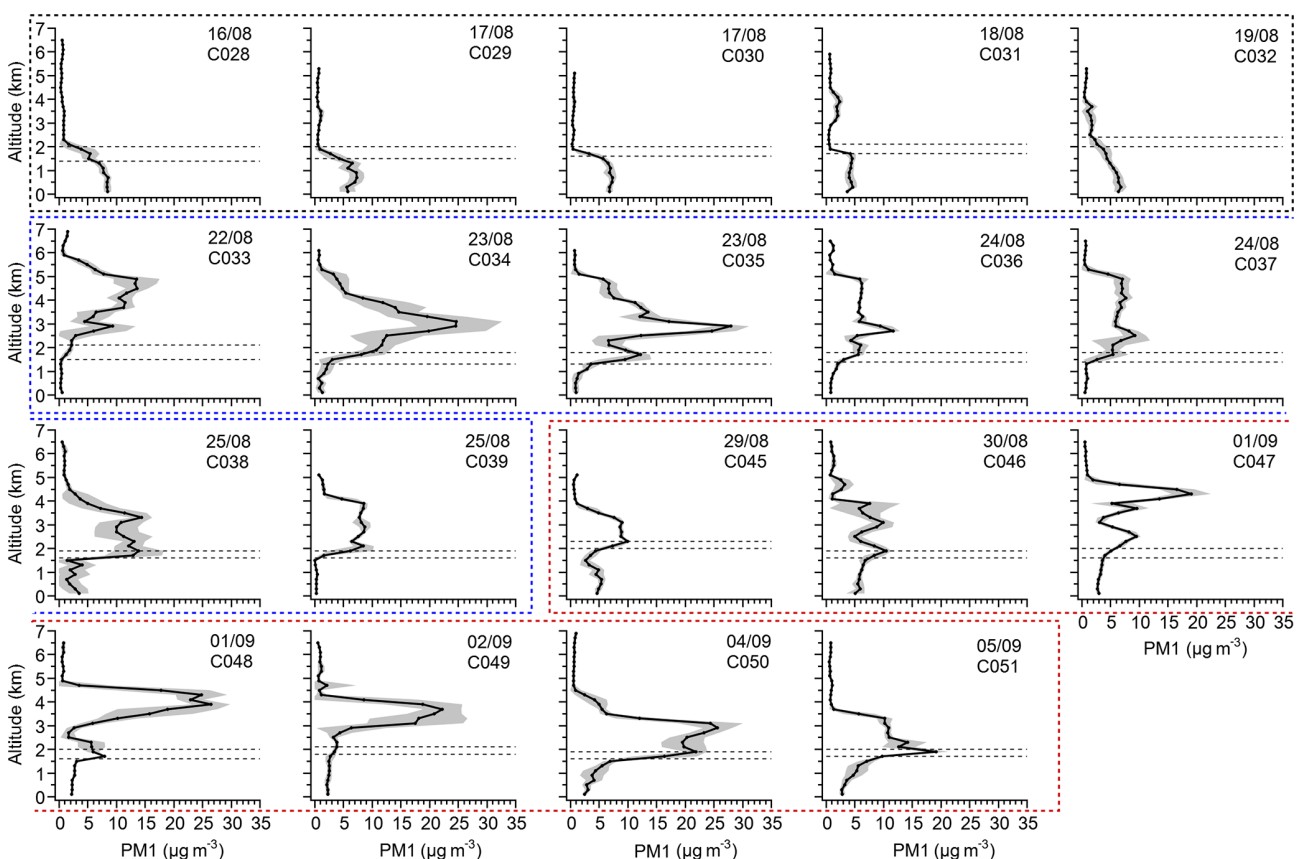

**Figure 2.** The average vertical distribution of submicron aerosol ($PM_1$) in flights used in this study. The mass concentration of $PM_1$ is calculated from AMS non-refractory submicron species and SP2 BC. The grey shades represent standard deviation. The dashed lines represent the low and upper level of the inversion layer. Period 1 is marked by the black dashed rectangle, which shows that BBA was concentrated in the BL. Period 2 is marked by blue, which shows that the FT was BB-polluted and the BL was mostly clean. Period 3 is marked by red, which shows that the BB pollution was observed throughout the BL and FT.

**Table 1.** The summary of CLARIFY aerosol composition and comparison with other studies of southern African BB.

|  |  | Origination | OA mass fraction | BC mass fraction | Nitrate mass fraction | Ammonium mass fraction | Sulfate mass fraction | Reference |
|---|---|---|---|---|---|---|---|---|
| CLARIFY BB-polluted FT | Period 2 | Transported | $63 \pm 5$ | $13 \pm 2$ | $8 \pm 3$ | $7 \pm 2$ | $11 \pm 3$ | This study |
|  | Period 3 | Transported | $60 \pm 5$ | $15 \pm 3$ | $8 \pm 3$ | $7 \pm 3$ | $12 \pm 4$ | This study |
| CLARIFY BB-polluted BL | Period 1 | Transported | $50 \pm 5$ | $8 \pm 2$ | $2 \pm 1$ | $10 \pm 2$ | $30 \pm 4$ | This study |
|  | Period 3 | Transported | $54 \pm 6$ | $14 \pm 2$ | $3 \pm 2$ | $8 \pm 3$ | $21 \pm 5$ | This study |
| CLARIFY clean BL[1] | Period 2 |  | $\sim 24$ | – | – | $\sim 16$ | $\sim 60$ | This study |
| SAFARI 2000, fresh plume |  | Near-source | 85 | 5 | 3 | 3 | 4 | Formenti et al. (2003)[2] |
| SAFARI 2000, aged plume (1–2 d) |  | Near-source | 71 | 6 | 6 | 5 | 12 | Formenti et al. (2003)[2] |
| DACCIWA West Africa, FT aged BB |  | Transported | $\sim 65$ | – | $\sim 10$ | $\sim 10$ | $\sim 15$ | Haslett et al. (2019b)[3] |
| DACCIWA West Africa, marine layer, aged BB |  | Transported | $\sim 58$ | – | $\sim 2$ | $\sim 10$ | $\sim 30$ | Haslett et al. (2019b)[3] |

[1] The nitrate and BC mass concentrations were around zero, have large uncertainty and are therefore not provided here. [2] Formenti et al. (2003) used a factor of 2 to convert measured organic carbon (OC) to organic mass. It should be noted that BC was not measured optically for blackness as SP2 in this study but instead for the elemental carbon (EC). [3] The fraction of BC is not provided by Haslett et al. (2019b); only data from the AMS are calculated, and hence the mass fractions of the non-BC components are likely elevated.

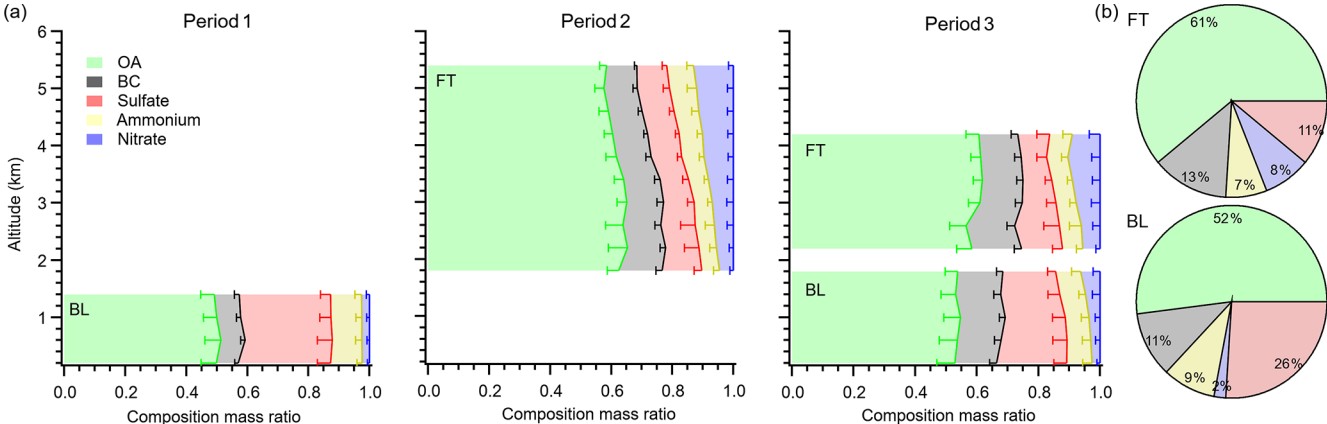

**Figure 3. (a)** The average vertical distribution of $PM_1$ chemical composition ratios in the BB-polluted FT and BL separately in each period. The width of colour bars represents the average mass ratio of different species in every 400 m bin. The error bars represent 1 standard deviation. **(b)** Pie charts showing campaign-average chemical composition ratios in the BB-polluted FT and BL respectively.

Table 1 also compares the chemical composition of BBA measured during CLARIFY and other studies focusing on southern African BB of different ages. The chemical composition of FT non-refractory BBA in CLARIFY is similar to the transported FT BBA in DACCIWA (Haslett et al., 2019b). During SAFARI 2000, off-line methods using filter samples were employed (Formenti et al., 2003). Concentration of water extractable ions ($NO_3^-$, $SO_4^{2-}$, $NH_4^+$) was determined using ion chromatography (IC), and total carbon (TC = organic carbon + elemental carbon) was determined using thermo-optical analysis techniques. The comparison shows composition differences between the fresh and aged BBA (1–2 d) in SAFARI 2000 and more aged BBA sampled during the CLARIFY and DACCIWA experiments. The lower OA fraction of more aged BBA is likely due to the possible OA loss after emission as a result of evaporation or oxidation (Hodshire et al., 2019), or the formation of the secondary inorganic components (Pratt et al., 2011).

We also observed vertical variation in the fractional chemical composition of the BB layers, as shown in Fig. 3. In the BB-polluted FT, the linear fitted C-ToF-AMS $NH_{4mea}^+/NH_{4neu}^+$ ratios of aerosols in period 2 and 3 were $(1.06 \pm 0.01)$ and $(1.05 \pm 0.02)$ respectively, indicating that sulfate was fully neutralized and nitrate aerosol was formed with the excess ammonia. Therefore, the amounts of measured nitrate, ammonium and sulfate reached ion balance in the FT. When the observed nitrate mass fraction increased with altitude (mean values ranged from 4 % to 13 % in period 2 and from 6 % to 11 % in period 3 respectively), the sulfate mass fraction was relatively constant, and the ammonium mass fraction consistently increased with altitude. The BC mass fraction generally decreased with altitude in the FT; mean values changed from 14 % to 9 % in period 2 and from 16 % to 12 % in period 3. In the BB-polluted BL, there was no significant vertical variability in period 1. In period 3, the sulfate mass fraction increased from 17 % at the top of the

BL to 23 % when close to the surface, while other component mass fractions showed slightly opposite trends (BC and nitrate mass fractions) or were relatively stable (OA and ammonium mass fractions).

### 3.1.2 Organic composition and elemental analysis

The OA fragment marker, $f60$, represents the prevalence of anhydrous sugars such as levoglucosan that are known pyrolysis products of wood burning. Hence, $f60$ is regarded as an indicator of emitted primary aerosol during BB (Schneider et al., 2006; Alfarra et al., 2007). Meanwhile, $f44$ is associated with the $CO_2^+$ ion and is a marker for oxidized OA (Aiken et al., 2008). The method used by Cubison et al. (2011) that relies on $f44$ vs. $f60$ to represent the ageing of BB OA in the atmosphere is reproduced in this work. This approach compares the increasing oxidation of the OA (increasing $f44$) with the oxidative decay of the levoglucosan-like species (decreasing $f60$), allowing a simplified description of BB OA ageing to be compared across different BB studies. Figure 4a shows the $f44$ vs. $f60$ diagram of the average values in each flight and compares these values with those obtained by previous studies. During CLARIFY, average $f60$ were calculated to be $(0.6 \pm 0.3)$ % and $(0.5 \pm 0.2)$ % in the BB-polluted FT and BL respectively. Previous field studies have sampled BBA from flaming fires at Lake McKay (Cubison et al., 2011) and Amazonia fires (Morgan et al., 2020). These previous studies observed much higher $f60$ at source and in the near-source region than that observed in this study. Substantial oxidation and loss of levoglucosan-like species has occurred in the CLARIFY region after > 7 d transport. Cubison et al. (2011) observed that $f60$ decayed to near background level (0.3 %, in air masses without BB influence) during 1 d transport. The $f60$ is currently thought to be a robust BB tracer for ageing timescales within 1 d from emission (Cubison et al., 2011; Ortega et al., 2013). However, Jolleys

et al. (2015) reported an average $f60$ of 1.2 % in aged BB smoke that had been transported $\sim 5$ d in the FT after emission from boreal forest fires, which is well above the 0.3 % background level, suggesting that the lofting of BB smoke into the FT may lead to the retention of levoglucosan-like species. The low values presented in this paper indicate $f60$ in the far-field region eventually decayed to near background levels even when the smoke was transported into the FT. Average $f44$ values from each flight were mainly in a range of 18 %–23 % and 20 %–25 % in the BB-polluted FT and BL. As shown in Fig. 4a, $f44$ and $f60$ values during CLARIFY lie in the left top of the panel, and $f44$ values are at a high level compared with other BB studies in source, near-source or transport regions (Cubison et al., 2011; Haslett et al., 2019b). These high $f44$ values indicate the large fraction of oxidized OA (OOA) and/or highly oxidized OA state.

We also calculated the elemental composition ratios of oxygen to carbon (O/C) and hydrogen to carbon (H/C) based on the estimates proposed by Aiken et al. (2008) and Ng et al. (2011). It should be noted that $f44$ in this study is at the top end of the $f44$ range reported by Aiken et al. (2008), and $f43$ is at the bottom end of the $f43$ range reported by Ng et al. (2011), and the aerosols were sampled from different fire sources in these studies; thus the O/C and H/C may have larger uncertainty than the reported error of 9 % (Aiken et al., 2008) and 10 % (Ng et al., 2011). The average carbon oxidation state (OSc) was estimated using O/C and H/C (Kroll et al., 2011). Figure 4b shows the Van Krevelen diagram (H/C vs. O/C), with average values in each flight and the boundaries of OSc in the BL and FT respectively, following the method in Ng et al. (2011). The elemental composition ratios and OSc are within the observed values of low-volatility oxygenated OA (LV-OOA) (Kroll et al., 2011; Ng et al., 2011). The organic-mass-to-organic-carbon ratios (OM/OC) (Aiken et al., 2008) were calculated as 2.1–2.4 in the FT and 2.2–2.5 in the BL, with the same median of 2.3. In general, ageing increases the oxidation state of OA, associated with increasing $f44$, O/C and OM/OC ratios (Jimenez et al., 2009). These high values consistently reflect the highly oxidized and low-volatility nature of BB OA in the CLARIFY region.

### 3.1.3 Enhancement ratios of BC and OA

The modified combustion efficiency (MCE) was calculated to indicate the combustion conditions at source (Yokelson et al., 2009). Details of the method of calculating MCE are listed in Sect. S1 in the Supplement. The MCEs of FT smoke were generally around 0.97 during CLARIFY, as shown in Fig. 5. An MCE > 0.9 is commonly used to indicate BB smoke predominantly influenced by combustion during the flaming phase, whereas MCE < 0.9 represents the smouldering phase (Reid et al., 2005). By this definition, CLARIFY smoke plumes transported from southern Africa are likely to be mostly controlled by flaming-phase combustion at source.

The emission of BC is usually high during flaming combustion, while smouldering combustion tends to emit smoke high in CO and organic mass (e.g. Christian et al., 2003). The enhancement ratios of BC and OA with respect to CO (BC/$\Delta$CO and OA/$\Delta$CO, $\mu$g m$^{-3}$ ($\mu$g m$^{-3}$)$^{-1}$) are generally used to indicate the emission conditions of fire at source. For example, BC/$\Delta$CO values from 0.005 to 0.023 and OA/$\Delta$CO values from 0.037 to 0.066 were observed for BB source in flaming combustion from previous measurements (May et al., 2014; Pratt et al., 2011), while a lower range of (0.0014–0.0072) for BC/$\Delta$CO and a higher range of (0.080–0.096) for OA/$\Delta$CO were reported for BB source in smouldering combustion (Capes et al., 2008; Kondo et al., 2011; May et al., 2014).

For CLARIFY, the BC/$\Delta$CO and OA/$\Delta$CO ratios ($\mu$g m$^{-3}$ ($\mu$g m$^{-3}$)$^{-1}$) were calculated in the FT by the unconstrained linear orthogonal distance regression (ODR) fit (Yokelson et al., 2013) and were calculated in the BL by dividing BC and OA by the excess concentration of CO, after background values had been removed (Lefer et al., 1994). The detailed calculation method is listed in Sect. S1. The calculated enhancement ratios in FT and BL smoke for each flight are shown in Fig. 5. In the BB-polluted FT, the ODR fitted BC/$\Delta$CO ratios ranged from 0.0087 to 0.0114 in period 2 and were higher (0.0103–0.0134) in period 3, while OA/$\Delta$CO values were comparable between the two periods (period 2: 0.042–0.067; period 3: 0.043–0.064). In the BB-polluted BL, the average BC/$\Delta$CO and OA/$\Delta$CO ratios in period 1 (0.0103–0.0111; 0.062–0.079) were higher than in period 3 (0.006–0.0085; 0.024–0.041). Particles are unlikely to have been subject to significant wet removal after being lofted into the FT, due to the low water contents and low probability of encountering clouds in the FT over the southeast Atlantic. Hence the FT aerosols are likely to be long-lived. It is also acknowledged that CO has a lifetime of around a month by gas-phase oxidation; this lifetime is much longer than the transport timescales in this study. Previous studies have observed the transatlantic transport of BB pollutants from Africa to the Amazon basin, reporting a BC/$\Delta$CO value of 0.0117 in FT transported smoke, which is within the observed range in this study (Baars et al., 2011; Holanda et al., 2020). For CLARIFY, it is likely that BC/$\Delta$CO values in FT smoke are similar to values at source. However, OA/$\Delta$CO may be more complex due to ageing of primary organics (POA) and secondary organic aerosol (SOA) formation after emission (Yokelson et al., 2009; Cubison et al., 2011; Vakkari et al., 2018). During CLARIFY, BC/$\Delta$CO ratios in FT smoke were in the reported range of BB sources controlled by flaming combustion. The slight variations between flights may be due to differences in emission if there is no significant removal process. Back trajectories initiated from Ascension Island (Zuidema et al., 2018; Haywood et al., 2020) and climate model simulations made by Gordon et al. (2018) indicated that the BB smoke from south African fires had entrained into the marine BL over the southeast At-

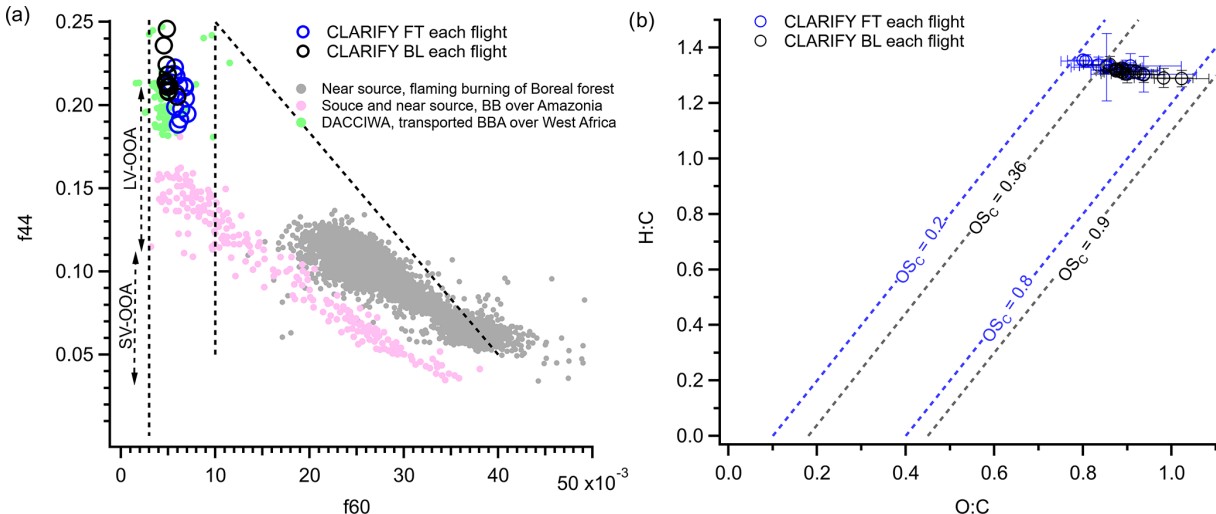

**Figure 4. (a)** The fractional signals $f44$ vs. $f60$ of BBA sampled in this and other studies. Blue and black markers represent the average of FT and BL BBA layers, respectively, for each flight. The vertical dashed grey line indicates the background of $f60$ (0.3 %) under non-BB conditions, as recommended by Cubison et al. (2011). **(b)** The average and standard deviation of H/C vs. O/C for sampled BBA in each flight and the boundaries of OSc in the BL (black) and FT (blue) respectively.

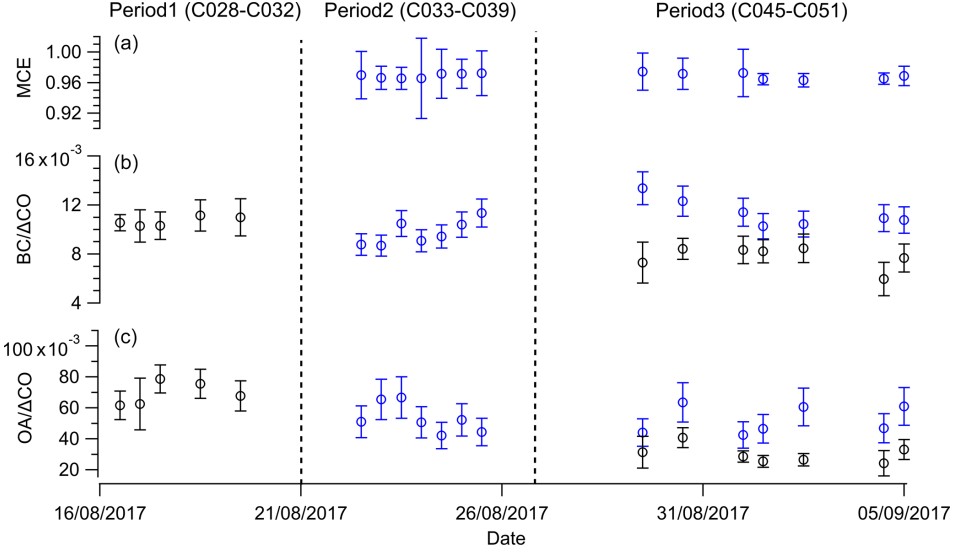

**Figure 5. (a)** The calculated MCE of CLARIFY FT smoke plumes for each flight. The error bars show the uncertainty. **(b, c)** The calculated BC/$\Delta$CO and OA/$\Delta$CO in FT and BL smoke plumes respectively for each flight. The blue markers and error bars represent the fitted slopes and uncertainty in the FT; the black markers and errors represent the average and standard deviation of calculated ratios in the BL.

lantic. BC/$\Delta$CO and OA/$\Delta$CO ratios in the BL were generally lower than that in the FT, clearly seen throughout period 3, indicating that a fraction of particles may be removed by cloud activation or scavenging and subsequent precipitation after FT aerosols mix into the BL. However, the ratios were not considerably lower than those in the FT, suggesting that the removal processes were inefficient. During period 3, the difference between BL and FT BC/$\Delta$CO varied from 20 % to 45 %, suggesting different scavenging fractions. In

the BL, both the differences due to emission and the extent of aerosol removal may cause the variation in the ratios.

## 3.2 Aerosol size distribution

We determined dry number size distributions from both PCASP and SMPS, during SLRs in the FT and BL separately. The mean size distributions of observed BBA from SLRs for each period are shown in Fig. 6a. During CLARIFY, we mainly detected a single dominant accumulation mode for

both FT and BL BBA. The lognormal fitted count median diameters (CMDs) of mean size distributions derived from the PCASP were 232 and 202 nm for the BB-polluted FT and BL respectively. Figure 6b shows the mean number size distribution for SLRs in the clean BL air masses in period 2. This indicates that new particle formation and growth was occurring in the background marine environment, with a CMD of $\sim 30$ nm in the Aitken mode and a CMD of $\sim 160$ nm in the accumulation mode.

Recent ground-based measurements of southern African savanna and grassland fires found a CMD of 69 nm in fresh smoke (age $< 0.5$ h), which grew to 123 nm in the next 3 h (Vakkari et al., 2018). CLARIFY-observed BBAs are much larger than those reported for fresh African smoke; this may be due to substantial coagulation and condensation during transport. However, Haywood et al. (2003a) reported a CMD of $\sim 240$ nm for aged BBA (1–2 d) off the Namibian coast and $\sim 200$ nm for fresh BBA ($\sim 5$ h) during SAFARI 2000. There is size similarity between SAFARI aged BBA (1–2 d) and more aged BBA ($> 7$ d) in this study, despite the different ages of aerosols. This consistency validates a priori size distribution assumptions for the aerosol model recently used in SEVIRI satellite retrievals of aerosols (CMD = 238 nm) made by Peers et al. (2019).

Vertical profiles of lognormal fitted CMDs calculated from the PCASP data are shown in Fig. 6c. There is a slightly increasing trend of CMDs with altitude (by $\sim 5\%$) in the BB-polluted FT and no significant vertical variability in the BL. CMDs in the BL were generally smaller than that in the FT, which is consistent with the lower BC/$\Delta$CO and OA/$\Delta$CO ratios in the BL than in the FT presented in the previous section and likely a result of more efficient removal of larger particles.

## 3.3 Aerosol single-scattering albedo

During CLARIFY, the column-weighted dry SSAs derived from EXSCALABAR measurements were 0.85, 0.84 and 0.83 at 405, 550 and 658 nm respectively. Vertical profiles of SSA were also calculated for each 400 m altitude bin. The profile for 658 nm is shown in Fig. 7 as an example; the same trends were observed at all reported wavelengths. In the BB-polluted FT, average SSAs at 405, 550 and 658 nm increased from 0.82, 0.81 and 0.79 in the low FT to 0.87, 0.86 and 0.85 at an altitude up to 5 km. In the BB-polluted BL, the SSAs decreased from the surface to the BL top. During period 3, average SSAs at 405, 550 and 658 nm were 0.85, 0.85 and 0.84 in the lowermost bin of 0–400 m, decreasing to 0.83, 0.81 and 0.80 at the BL top. The BL SSAs in period 1 showed a weak vertical change and were higher than in period 3.

Figure 8 shows the CLARIFY average SSAs at different wavelengths in the BL and FT separately, compared with previous observation studies of southern African BB at different ages and covering various relevant regions. In the source region, the average SSAs of fresh BBA measured during SA-FARI 2000 were 0.86, 0.84, and 0.80 at 450, 550 and 700 nm; aged BBA (1–2 d) usually had higher SSAs (Haywood et al., 2003a, b; Johnson et al., 2008). However, SSAs of more aged BBA ($> 4$ d, mainly in the FT) during ORACLES 2016 were observed to be lower than the SAFARI aged BBA (1–2 d) (Pistone et al., 2019). The CLARIFY observations presented in this study were made further west than the ORACLES region and had undergone additional days of ageing ($> 7$ d since emission). The average SSAs of CLARIFY FT BBA were $(0.85 \pm 0.02)$, $(0.83 \pm 0.03)$ and $(0.82 \pm 0.03)$ at 405, 550 and 658 nm respectively, falling within the lowest level of the above-reported range. The average SSAs of CLARIFY BL BBA were $(0.86 \pm 0.02)$, $(0.85 \pm 0.03)$ and $(0.84 \pm 0.03)$ at 405, 550 and 658 nm respectively, which is higher than FT values. Ground-based in situ SSA measurements made on Ascension Island in 2017 (Zuidema et al., 2018) are lower than CLARIFY BL SSA values and are the lowest values compared to all previously reported observations of southern African BBA.

These previous observations employed filter-based measurements, using the particle soot absorption photometer (PSAP) and nephelometer, in contrast to the PAS/CRDS methods employed during CLARIFY. CLARIFY FT SSA values are similar to those measured from the FT during the ORACLES mission. It is also interesting to note that the radiometrically retrieved SSA from nine above-cloud flights performed during ORACLES in 2016 and 2017 (Cochrane et al., 2020; their Fig. 4), which do not depend on in situ measurements, yielded average SSAs of $(0.85 \pm 0.02)$, $(0.83 \pm 0.03)$ and $(0.82 \pm 0.04)$ at wavelengths of 380, 550 and 660 nm respectively for FT BBA. These values are also in good agreement with our FT SSAs within the expected variability. However, CLARIFY BL SSA values do not agree with those from LASIC ground-based measurements. Although limitations with filter-based measurements of aerosol light absorption are known to introduce systematic measurement biases (Lack et al., 2008; Davies et al., 2019), the LASIC-derived aerosol absorption is comparable with those from the CLARIFY campaign. The difference between CLARIFY and LASIC BL SSAs is possibly due to differences in the extinction measurements, which may be caused by the different inlet cut-offs (aerosol dynamic diameter of 1 µm for LASIC and 1.3 µm for CLARIFY).

Despite the systematic variability between different measurement methods, the datasets mentioned above imply some important information on SSA evolution from the African BB source to the remote region. Abel et al. (2003) showed that SSA increased in the first 5 h after emission during SA-FARI 2000, which is likely due to the condensation of scattering material and the change in BC morphology from a chain agglomerate to a more spheroidal shape because the particle collapses as it becomes coated. Despite this initial increase, observations of SSA in regions where the aerosols are highly aged ($> 4$ d since emission), like the ORACLES, CLARIFY and ground-based measurements on Ascension

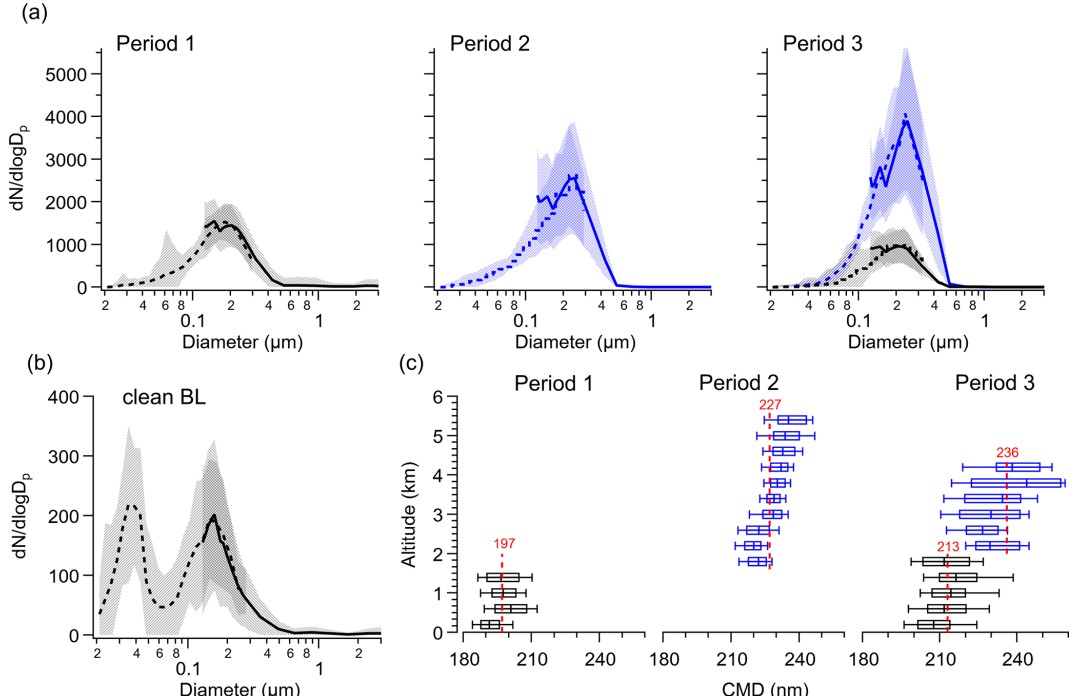

**Figure 6. (a)** The average size distributions of SLRs in the BB-polluted FT (blue) and BL (black) respectively for each period. The solid lines represent results from PCASP; dashed lines represent results from SMPS. The blue lines and shading show mean and standard deviation from the FT; the black lines represent the BL. **(b)** The average size distribution of SLRs in the clean BL. **(c)** The vertical distribution of lognormal fitted count median diameters (CMD) from the PCASP. The box-and-whisker plots indicate the 10th percentile, 25th percentile, median, 75th percentile and 90th percentile in every 400 m bin in the BB-polluted FT (blue) and BL (black). The red dashed lines and numbers represent the lognormal fitted CMD of mean size distribution in the BB-polluted FT (blue) and BL (black) for each period.

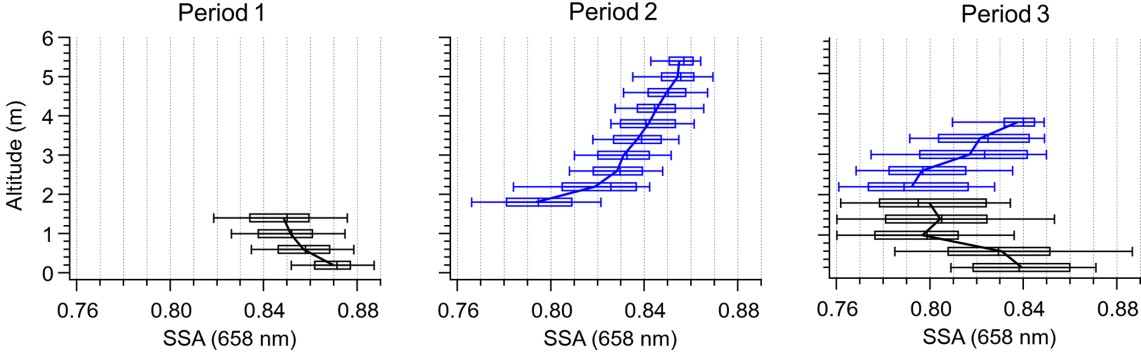

**Figure 7.** The vertical distribution of SSA at 658 nm in the BB-polluted FT (blue) and BL (black) respectively for each period. The box-and-whisker plots represent the 10th percentile, 25th percentile, median, 75th percentile and 90th percentile in every 400 m bin. The lines are the trend of average values in every 400 m bin. TS4

Island, are close to or lower than those sampled closer to source (< 2 d). These observations show that BBA remains strongly absorbing from near the coast of southern Africa to the far-field region around Ascension Island, suggesting that models with too little absorption for aged BBA will underestimate the warming effect of BBA over the southeast Atlantic.

## 4 Discussion

### 4.1 Factors influencing vertical variability

#### 4.1.1 In the FT

CLARIFY OA was highly oxidized, which is characteristic of aged, low-volatility organic aerosol. Aerosol properties will be relatively insensitive to further ageing pro-

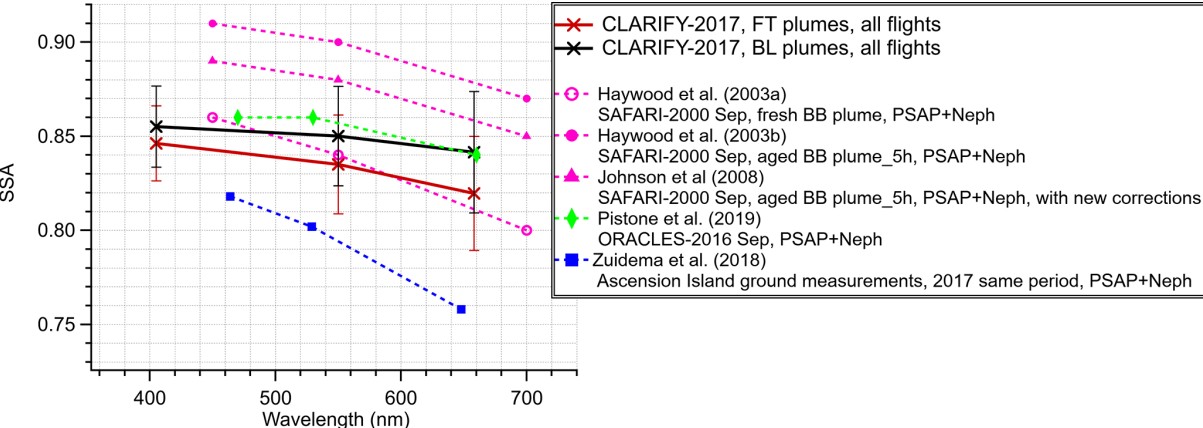

**Figure 8.** Wavelength dependence of the average SSA of FT and BL BBA for all flights used in this study. The markers and lines represent the mean value and standard deviation. The average SSAs from previous studies in this region are shown for comparison. (Note that the PSAP only measured absorption at 567 nm in SAFARI 2000; assumptions about the wavelength dependence of absorption coefficient were made to estimate absorption at 450 and 700 nm, which was then used to calculate the SSA.)

cesses of OA. The main feature in the vertical variability of aerosol properties in the CLARIFY region is the nitrate aerosol which makes a greater fractional contribution to $PM_1$ at higher altitudes.

Individual aerosol layers at different altitudes may have different source or transport history, as evidenced by the back-trajectory studies in Haywood et al. (2020), very probably leading to variation in the fractional chemical composition. CLARIFY measurements show that the nitrate aerosol was largely inorganic and existed in the form of ammonium nitrate ($NH_4NO_3$) in the FT (see Sect. S2), which is a semi-volatile and hygroscopic inorganic salt. The increasing nitrate mass fraction with altitude could also be reasonably explained by the chemical thermodynamics of the $HNO_3$–$NH_3$–$NH_4NO_3$ system across large temperature gradients (temperature could drop over 20 K from the low FT to the top of the aerosol layer, as seen in Fig. 9). During some flights, individual layers were well mixed, indicated by a constant potential temperature throughout their depth. In these smoke layers, increasing mass concentrations of nitrate and ammonium with increasing altitude were observed, while other species were relatively invariant with altitude. An example of a well-mixed smoke layer from flight C036 is shown in Fig. 10a and b. We conducted a simulation of the chemical thermodynamics in this example smoke layer, using a temperature-dependent thermodynamic model described in Friese and Ebel (2010). The inputs of ambient conditions (temperature and water content) and inorganic compositions (sulfate, nitrate and ammonium) were set to measured values in flight C036, and the ammonia value was assumed based on previous savanna wildfire studies in Andreae (2019). With the same initial compositions, the nitrate and ammonium concentrations were simulated over a measured temperature range (at different altitudes) from 281 to 269 K. The modelled nitrate and ammonium showed an increasing trend with

height, in a similar way to that of the measurements (see Fig. 10c), suggesting lower temperatures at higher altitudes would shift the gas–particle partitioning of the $HNO_3$–$NH_3$–$NH_4NO_3$ system toward the aerosol phase and significantly increase the amount of $NH_4NO_3$. The intrusion of BB smoke in the FT during periods 2 and 3 increased specific humidity compared with the cleaner FT in period 1 (see Fig. 9), since the FT smoke tends to coexist with enhanced water vapour as discussed in Adebiyi et al. (2015). With relatively constant specific humidity in BB smoke over the vertical profile, the measured and simulated RH (Figs. 9 and 10c) were both shown to increase at higher altitudes, consistent with colder temperatures aloft. As RH values reach 70 %, i.e. at the top of the aerosol layers around 5 km, aerosols are likely to become liquid particles and allow $NH_4NO_3$ to dissolve in the aqueous aerosol phase. In summary, there is a greater chance for nitrate to be present in the aerosol phase in the colder and higher RH atmosphere encountered towards the top of the aerosol layers.

With higher nitrate mass fraction at higher altitudes, BC constituted a smaller mass fraction, while the BC number fraction remained relatively constant in the FT (see Fig. 11). This indicates that the additional nitrate is likely to be mostly internally mixed with existing particles. From the low FT up to 5 km, the mass was observed to increase by $\sim 15$ % due to the additional $NH_4NO_3$. The average aerosol composition fractions in the low FT were observed to be 64 %, 14 %, 6 %, 12 % and 4 % for OA, BC, ammonium, sulfate and nitrate. The average density of a particle in the low FT was estimated to be 1.356 g cm$^{-3}$ following the method in Haslett et al. (2019a), assuming all particles are internally mixed. The density of $NH_4NO_3$ is assumed to be 1.725 g cm$^{-3}$ (Haslett et al., 2019a). When the additional $NH_4NO_3$ is internally mixed, it is estimated to lead to a $\sim 4$ % increase in aerosol radius, assuming the particles are spherical. This is consis-

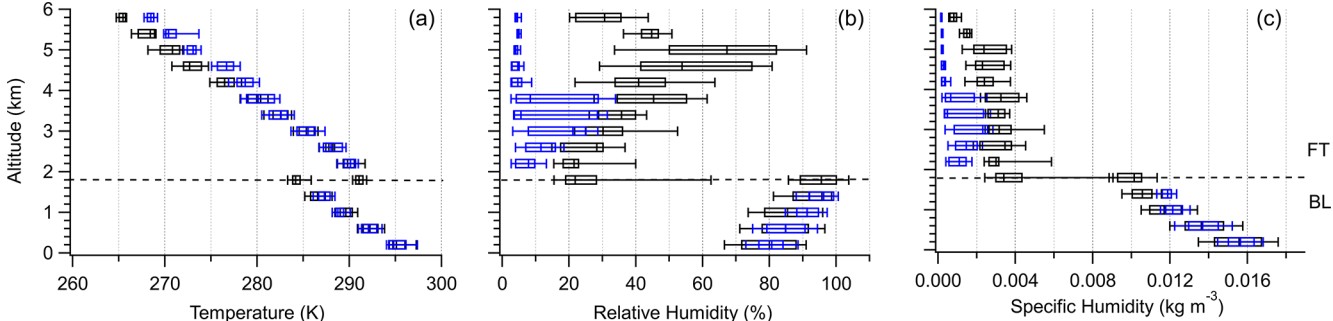

**Figure 9.** The vertical distributions of temperature **(a)**, RH **(b)** and specific humidity **(c)** under clean (blue) and BB-polluted (black) conditions. Data for BB-polluted conditions are composited from periods 2 and 3 in the FT and periods 1 and 3 in the BL. Data for clean conditions are composited from period 1 in the FT and period 2 in the BL. The box-and-whisker plots represent the 10th percentile, 25th percentile, median, 75th percentile and 90th percentile in every 400 m bin.

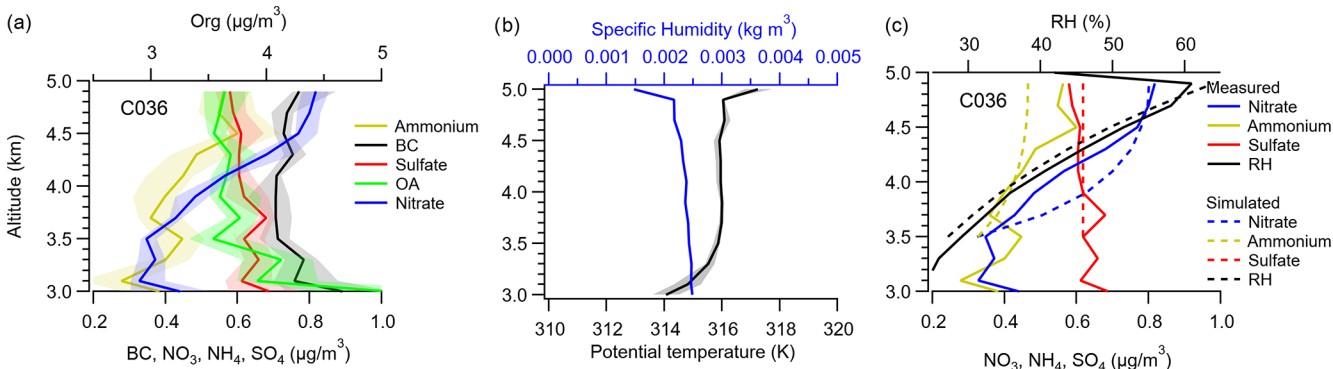

**Figure 10.** The vertical distributions (3000–5000 m) of **(a)** different chemical composition concentrations and **(b)** potential temperature and specific humidity in flight C036 (24 August). The lines and shades represent the 25th percentile, median and 75th percentile in every 200 m bin. **(c)** The simulated and measured chemical composition and RH at different altitudes with variable temperatures.

tent with the slight vertical change of CMDs of bulk aerosols in the FT. It is likely that this internal mixing did not significantly alter the overall dry aerosol size distributions. SSA is closely related to the particle size and chemical composition. The slightly increased particle size and the larger fraction of scattering material at higher levels would consistently contribute to the increasing SSA with altitude observed during CLARIFY.

In this study, the calculated SSAs from the PAS and CRDS instruments are for dry aerosols. It is well known that an increase in RH can result in an increase in aerosol scattering, since particle size and refractive index vary with particle water content (e.g. Zieger et al., 2013; Burgos et al., 2019). In the CLARIFY region, increasing RH with altitude in the FT is likely to result in an increase in aerosol size and scattering, when aerosol particles are most likely to acquire water near the top of the aerosol layers. Previous studies have reported that aerosol absorption can be also affected by humidification. However, it is noted that most of studies considering the effect of humidification on aerosol absorption observed little or no increase in absorption for RH < 85 % (e.g. Brem et al., 2012). The RH of observed smoke in the FT during CLAR-

IFY was rarely over 80 %. If there is little effect of humidity on absorption, we would expect that the impact of humidification is likely to increase SSA at higher levels, indicating a substantially larger vertical variation in SSA in the FT.

### 4.1.2 In the BL

The entrainment of FT smoke is a recognized source for BL BBA over the southeast Atlantic (Gordon et al., 2018; Zuidema et al., 2018; Haslett et al., 2019b). There are two important factors that are likely to alter aerosol properties after FT BBA mix into the BL. The first factor is marine emissions in the BL, and the second is removal processes as evidenced by lower BC/$\Delta$CO and OA/$\Delta$CO ratios in the BL than in the FT.

Dimethyl sulfide (DMS) from oceanic biogenic emission is an important source of sulfate precursor, $SO_2$, and sulfate aerosol (Perraud et al., 2015). The clean BL described in Sect. 3 suggests new particle formation and growth and a marine sulfate background. Some of these marine sulfates would become internally mixed with BBA either by condensation of $H_2SO_4$ or by cloud processing, thus driving nitrate

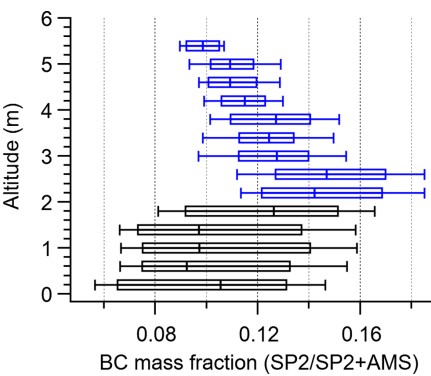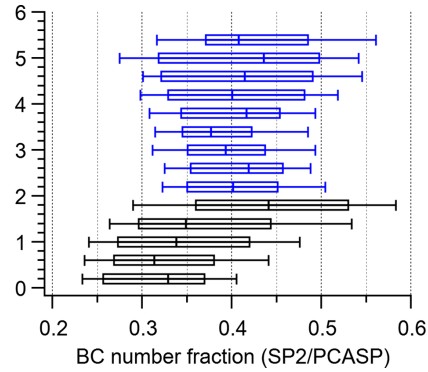

**Figure 11.** The vertical distributions of BC mass fraction and number fraction in the BB-polluted FT (blue) and BL (black) respectively. The box-and-whisker plots represent the 10th percentile, 25th percentile, median, 75th percentile and 90th percentile in every 400 m.

to evaporate into the gas phase and causing the loss of nitrate aerosol in the BL. Taylor et al. (2020) did not observe thicker BC coatings in the BL than those in the low FT, while in this study the sulfate mass fraction in the BL was significantly enhanced, suggesting that some of the marine sulfate would be also externally mixed with BBA. Sea salt particles from sea spray can also provide submicron particles which exhibit an SSA close to 1. The mixing of more scattering material from marine emissions can result in a higher SSA in the BL than in the FT.

The wet removal events usually occur via aerosol activation to form cloud droplets during in-cloud processing and subsequent removal of those droplets by precipitation, which would also facilitate below-cloud aerosol scavenging (Moteki et al., 2012; Taylor et al., 2014). In this study, OA dominated the aerosol composition and was characterised by high $f44$, which is closely associated with carboxylic acid content (Heald et al., 2010; Duplissy et al., 2011). The aerosols with a large proportion of inorganic species and OA are likely to be hygroscopic. Larger aerosol particles which are hygroscopic were preferentially activated and scavenged during removal events in the BL; thus the dry CMDs of remaining bulk aerosols in the BL were smaller than those in the FT. However, the removal rates were not as significant as previous studies of BBA removal affected by strong precipitation events which show a scavenging fraction of over 80 % (Taylor et al., 2014). This suggests that the scavenging efficiency of removal by drizzle in the marine BL was not large in the CLARIFY region. Our measurements show that the extent of this removal process is sufficient to reduce the dry CMD by ∼ 10 % (Fig. 6).

The BC mass and number fractions in the BL were both lower when close to the surface (see Fig. 11). This may suggest variations in the extent of external mixing between BBA and marine particles throughout the BL. In the CLARIFY region, the widespread stratocumulus clouds commonly lead to a decoupled BL (Lock et al., 2000; Gordon et al., 2018). Abel et al. (2020) showed an example structure of decou-

pled BL during period 3, with an unstable layer from the sea surface up to an altitude of about 600 to 700 m and then another layer up to the main BL inversion. The surface layer is likely to have a significant source of marine sulfate from secondary formation as well as submicron sea salt aerosol from sea spray. This could explain the higher sulfate mass fraction and higher SSA close to the sea surface.

These properties (chemical, size and optical) and variations that we have reported are all for dry aerosols in the BL. The RH in the BB-polluted BL was mostly over 80 % and up to 95 % at the BL top (Fig. 9), which would result in significant aerosol growth and scattering enhancement. Based on a scattering enhancement factor of ∼ 1.4 at RH of 80 % reported for ORACLES (Pistone et al., 2019) and SAFARI (Magi et al., 2003) aged BBA, there will be an increase in SSA by 0.03–0.05 in the BL, without considering absorption change. However, in reality, this value will be lower since absorption enhancement is suggested to be significant at high humidity (RH > 85 %) (e.g. Brem et al., 2012), which would have the opposite effect of lowering SSA. Due to high uncertainties surrounding these competing effects, more quantification studies of humidification impacts on aerosol optical properties are needed to determine BL biomass burning SSA in detail in this region.

### 4.2 Drivers of the low SSA

Previous measurements of fresh or transported BBA from forest fires in the Amazon, Siberia and North America reported a range of (2 %–9 %) for the average BC mass fractions of BBA (Kondo et al., 2011; Sahu et al., 2012; Artaxo et al., 2013; Allan et al., 2014; Morgan et al., 2020). Corresponding average dry SSA (at ∼ 550 nm) ranged from 0.88 to 0.97, using in situ measurements with the PSAP and nephelometer (Corr et al., 2012; Johnson et al., 2016; Laing et al., 2016). Compared with other BB-type regions, BBA during CLARIFY was richer in BC, with larger BC mass fraction and lower SSA. Many factors contribute to the larger BC mass fraction. Burning sources of the CLARIFY transported

BB smoke were controlled by flaming combustion with very high MCE, indicating that the emission of BC is likely to be proportionally high. The burning fuel of the southern Africa savanna is also suggested to have higher BC emission factors than forests or peat (Andreae, 2019). Both high MCE and the fuel type would lead to BC-rich smoke plumes at sources. OA loss induced by the ageing process and volatilization of semi-volatile material during dilution are likely to further enhance the BC mass fraction. This is consistent with the chemical composition comparison between BBA of different ages from other relevant studies, detailed in Table 1. In the absence of significant removal over the South Atlantic, these BC-rich smoke plumes from the southern Africa fires lead to the high BC contents far offshore, even after > 7 d transport. The high BC fraction leads to a large fraction of absorbing material in the sampled BBA and therefore contributes to the low observed SSA during CLARIFY.

The mass absorption cross section (MAC $= B_{Abs}$ / BC mass concentration) describes the absorption efficiency of BC particles. In the CLARIFY region, MAC was found to be much higher than the MAC of fresh, uncoated BC (MAC $= 7.5 \, \mathrm{m^2 \, g^{-1}}$ at 550 nm) suggested by Bond and Bergstrom (2006). Average MAC values from ground-based measurements at Ascension Island were reported as 15.1, 13.3 and 10.7 $\mathrm{m^2 \, g^{-1}}$ at 464, 529 and 648 nm respectively (Zuidema et al., 2018). Higher MAC values (20.3, 14.6 and 11.8 $\mathrm{m^2 \, g^{-1}}$ at 405, 514 and 658 nm respectively) were also observed from aircraft observations during CLARIFY (Taylor et al., 2020). The absorption Ångström exponent (AAE) of CLARIFY BBA was reported to be close to 1 (Taylor et al., 2020). It is assumed that an AAE over 1 indicates absorption from particles like brown carbon (BrC) or dust which have higher AAEs than BC (Lack and Langridge, 2013). There was only a minor non-BC material contribution to the total aerosol absorption in the CLARIFY region (Taylor et al., 2020). The enhanced absorption is therefore likely to be mostly due to the observed thick coatings on BC (Taylor et al., 2020), causing a lensing effect and additional absorption of sunlight (Lack et al., 2009). The high MAC values of BC would also contribute to the relatively low observed SSA.

The relatively low dry SSA measured during CLARIFY, as determined by highly sensitive and accurate measurements that are not subject to the artefacts of filter-based methods, is an important result. The SSA of aged BBA used in climate models is generally higher than the SSA in this study (e.g. Randles and Ramaswamy, 2010; Johnson et al., 2016; Herbert et al., 2020). Furthermore, the vertical profiles of SSA show that the lowest values (averages: 0.82, 0.81 and 0.79 at 405, 550 and 658 nm) occur at low FT layers around 2000 m altitude, immediately above the stratiform cloud. The air is also relatively dry within these low FT layers, meaning that the measured dry SSA is analogous to ambient condition. This is important as the positive radiative feedback associated with the aerosol direct effects may be underestimated in current models, especially for the cases with low and thin

smoke layers above clouds. Herbert et al. (2020) also found that both the cloud response and semi-direct radiative effects increase for thinner and denser overlying aerosol layers with lower SSA. The bias in modelled SSA values is likely to lead to misrepresentation of semi-direct effects as may neglecting the vertical variation in SSA. These findings suggest that modelled climate effects of BBA in this region need reassessment in future studies, and the variation in SSA values in different BB regions should be considered.

## 5 Conclusions

We have presented a detailed study of BBA chemical, physical and optical properties from the CLARIFY aircraft campaign, based from Ascension Island in the southeast Atlantic Ocean. These are the first accurate in situ airborne measurements providing aerosol vertical information in this area, which is affected by long-range transport of southern African BBA every year and is important climatically. Our dataset complements previous observations of the southern African BBA and extends previous studies to a wider geographical range and to a greater age of smoke. It provides unique parameterizations with which to constrain global and regional climate models and predict radiative effects across this region.

BB smoke plumes during CLARIFY have been shown to be mostly controlled by flaming combustion at their sources, and BBA has not undergone significant removal processes before arrival in the CLARIFY region, since enhancement ratios of BC remain relatively high. Transported submicron BBA was mainly composed of OA (50 %–60 % by mass) and BC (8 %– 15 % by mass), over the southeast Atlantic. The particles have undergone a significant ageing process during > 7 d transit from source, as indicated by highly oxidized and low-volatility OA in this study and thickly coated BC in Taylor et al. (2020). CLARIFY data provide a good representation of highly aged aerosols from the southern African BB.

The highly aged BBA in the CLARIFY region has relatively low dry SSA as the BBAs are rich in BC and the MAC of the sampled BC is high. The column-weighted dry SSAs were observed to be 0.85, 0.84 and 0.83 at 405, 550 and 658 nm respectively. We also observed vertical variability of the dry SSA: the lowest SSA (averages: 0.82, 0.81 and 0.79 at 405, 550 and 658 nm) in the column was generally in the low FT layer around 2000 m altitude, and the SSA increased with altitude in the FT. In the BL, the SSA decreased from the surface to the BL top, with the highest SSA in the column observed in the near-surface layer. The measured BBA in the CLARIFY region is generally more absorbing than currently represented in climate models. Considering these BBAs have a long lifetime and their spatial range spans thousands of kilometres, and the direct and semi-direct radiative effects of smoke layers in the southeast Atlantic area are highly sensi-

tive to the absorbing properties of BBA (Mallet et al., 2020), modelled climate effects need reassessment over this region.

In the CLARIFY region, the observed vertical variation in SSA is likely to be a persistent feature, which is a function of vertical variations in relative chemical composition, size and mixing state of these aerosols. In the FT, the main driver for vertical variability is the thermodynamic processing of inorganic nitrate driven by lower temperatures and higher RH at the top of the BBA layer. The increasing fraction of condensed nitrates is likely to be internally mixed with existing particles, which alters the relative chemical composition but does not significantly change the aerosol size distributions. Increases in the dry SSA with altitude are associated with the larger fractions of scattering material and slightly increased particle size at higher levels. These effects describe the variation in the dry aerosol properties. However, considering the effect of elevated RH on aerosol scattering at higher altitudes, the vertical variation in SSA is likely to be more significant when adjusted to ambient conditions.

The aerosols in the BL are essentially separate from the FT. Once aerosols are entrained into the BL, the BBA circulates independently of the aerosol above it owing to the strong inversion. There are two important factors affecting aerosol properties in the BL. One is marine emissions providing marine sulfate and sea salt, which can be internally or externally mixed with BBA. Another one is the possible aerosol removal by drizzle, resulting in smaller bulk aerosol size distributions. A larger fraction of scattering material may lead to a higher average dry SSA in the BL than in the FT. Vertical variability of aerosol properties exists since the BL is commonly decoupled over the southeast Atlantic. A larger concentration of marine sulfate or submicron sea salt is more likely to be present in the surface layer than above, leading to more scattering material and therefore higher SSA.

These observations provide new information in a climatically important region and demonstrate that the persistence of strongly absorbing aerosol from southern African BB across wide regions of the South Atlantic is prevalent and must be taken into account when considering regional radiation interactions. The observed vertical variation in aerosol properties throughout the BL and FT, especially SSA, should be also considered as part of any future studies which rely on prescribed aerosol composition and optical properties.

*Data availability.* Airborne measurements are available from the Centre for Environmental Data Analysis (https://catalogue.ceda.ac.uk/uuid/38ab7089781a4560b067dd6c20af3769, Facility for Airborne Atmospheric Measurements et al., 2017).

*Supplement.* The supplement related to this article is available online at: https://doi.org/10.5194/acp-20-1-2020-supplement.

*Author contributions.* HC and JH designed the research; JWT, JML, PIW, MF, JP, MIC, SJA, CF and NWD performed field experiments; HW, JWT, KS, JML, JDA and JP prepared datasets of AMS, SP2, PAS, CRD and FGGA; HW and JWT analysed datasets; and HW, JWT and HC wrote the paper.

*Competing interests.* The authors declare that they have no conflict of interest.

*Special issue statement.* This article is part of the special issue "New observations and related modelling studies of the aerosol–cloud–climate system in the Southeast Atlantic and southern Africa regions (ACP/AMT inter-journal SI)". It is not associated with a conference.

*Acknowledgements.* The staff of Airtask, Avalon Engineering and the Facility for Airborne Atmospheric Measurements (FAAM) are thanked for their thoroughly professional work, before, during and after the deployment.

*Financial support.* This research has been supported by the Natural Environment Research Council (grant no. NE/L013584/1).

*Review statement.* This paper was edited by Paquita Zuidema and reviewed by Steven Howell, Allison C. Aiken, and one anonymous referee.

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

**Remarks from the typesetter**

**TS1** Please note the that figures in the original submission have been adjusted to our standards. Note that we have only made the unit changes (sm to m in Fig. 2) to the preexisting figure and no other figures have been replaced. To use the new figures you provided, please explain the changes made to each figure as well as reasoning for why these changes are necessary since material changes require editor approval before being inserted. Please also clarify what you mean by "changes to composition numbers" in the new figures.

**TS2** According to our standards, changes like this must first be approved by the editor, as data have already been reviewed, discussed and approved. Please provide a detailed explanation for those changes that can be forwarded to the editor. Please note that this entire process will be available online after publication. Upon approval, we will make the appropriate changes. Thank you for your understanding. Note that we have inserted the changes to standard meters, but as the change from g $\text{kg}^{-1}$ to kg $\text{kg}^{-1}$ would change the value magnitude, these changes require prior approval.