# Peer review of "Vertical variability of the properties of highly aged biomass burning aerosol transported over the southeast Atlantic during CLARIFY-2017"

_Atmospheric Chemistry and Physics, 2020_

## Referee Comment (RC1) · Allison C. Aiken (Referee) · 11 May 2020

Manuscript No.: acp-2020-197

Title: Vertical variability of the properties of highly aged biomass burning aerosol transported over the southeast Atlantic during CLARIFY-2017

General Comments The authors present vertical information on long-range transport biomass burning (BB) aerosol from Southeastern Africa as sampled from aircraft measurements during CLARIFY-2017. The paper is focused on aerosol optical properties, chemical composition, size distributions, and emission factors. The work is thorough

and of great interest in understanding how BB aerosol ages post emission within the boundary layer and free troposphere, especially the direct vertical profile measurements of single scattering albedo and chemical composition. This area of the globe is lacking detailed measurements such as the ones the authors present here, which are the first of their kind and done in a well-organized format. The information on the vertical structure will be very informative for understanding BB aging and aerosol-cloud interactions for this remote region of the globe to reduce the climate model uncertainties that are very high in this region. I have some comments and suggestions that are focused on improving the impact as well as some technical comments that should be addressed before final publication within the ACP Special Issue: New observations and related modelling studies of the aerosol-cloud-climate system in the Southeast Atlantic and southern Africa regions (ACP/AMT inter-journal SI). Overall, I support this work for publication in ACP.

**Detailed Comments**

1 – Page 1, Line 27 – Reporting the lowest SSA values in the FT and the location at 2 km is very noteworthy since as the authors mention, this means the BB aerosol in the region is more absorbing that what is currently used in climate models. Would it be possible to report an average column-weighted SSA as well? This would be useful for comparison with satellite data, other passive ground sampling and modeling efforts. If so, this could also be reported for the BL and FT separately as well in the following sentences. If not, then the ranges should be included for the BL and FT.

2 - Page 1, Line 30 - Recommend stating whether the SSA is highest in the BL or highin the FT since the earlier sentences say the SSA increases in the FT with altitude soit is not clear in the abstract where the highest SSA was observed.

3 – Page 2, Line 54 - What about the other effects on clouds besides CCN formation? Later on, two paragraphs following, the same paper is referenced having calculated direct, indirect and semi-direct effects. It seems disjointed to not mention these effects
earlier.

4 – Page 2, Lines 54-55 – Another more recent publication also found significant enhanced convection over the Amazon region due to particles

10 - Page 5, Line 134 - 135 - Did the composition dependent CE that was determined based on Middlebrook et al. 2012 for the AMS data here include a mass comparison to the SMPS to validate the CE used?

11 – Page 5, Line 137 – Suggest including references to the early f44 and f60 work by Cubison et al., ACP, 2011 (already in your references), Ng et al., ACP, 2010 (https://doi.org/10.5194/acp-10-4625-2010), and Ortega et al., 2013 (doi:10.5194/acp-13-11551-2013) here where these factors are first mentioned in the text.

12 – Page 6, Lines 193 – 194 – How many flights were used out of how many total to make the temperature and specific humidity profiles? This information should be added to the main text and listed in the SI Figure caption.

13 - Page 7, Lines 199 - 200 - Can you state in the text how thick the typical inversion layer is between the BL and FT?

14 – Page 7, Line 225 – The higher SO4 and lower BC are more striking than the 10% change in OA fraction. State here the % changes in these species for the BB-polluted BL Period 1 vs 3 comparison.

15 – Page 8, Line 227 - With as much as the mass fractions are mentioned in the text, a bar or pie chart with the % mass fractions indicated would be a good visual representation of this comparison that could be added to Figure 3.

16 – Page 8, Lines 226 – 227 – State the sulfate fractions in the BL and FT for completeness and a quick comparison in the text so the reader doesn't have to go to the Figures/Tables to get this information.

17 – Page 8, Lines 247 – 249 - This is an interesting topic that deserves more attention. A recent publication reviewed lab and field BBA aging of OA and should be included in this discussion is presented by Hodshire et al., ES&T, 2019 (https://doi.org/10.1021/acs.est.9b02588). How common is the observation of secondary inorganic aerosol formation due to aging in ambient and/or laboratory data?

**ACPD**
Is this unique to BBA?

18 – Page 9, Line 273 – Consider adding Ortega et al., ACP, 2013 (doi:10.5194/acp-13-11551-2013) here in addition to Cubison et al., ACP, 2011.

19 – Page 9, Line 275 – Is the  $\sim$ 5 days quoted the timeline of the less aged or more aged BBA? It is not clear in the way the sentence is currently written. Also, how does this relate to the earlier statement that mentions f60 is only a good tracer for BBA with aging timescales

26 - Page 14, Lines 420 - 421 - Does "removal processes" include wet and dry deposition?

27 – Page 14, Line 421 – Same as earlier comment – suggestion to restate to include "new particle formation and growth".

28 – Page 14, Line 433 – Consider adding Heald et al., GRL, 2010 when referencing carboxylic acid content in addition to Duplissy et al., 2011). - https://agupubs.onlinelibrary.wiley.com/doi/full/10.1029/2010GL042737

29 - Page 15, Lines 475 - 477 - State why enhanced MAC's are being attributed to coatings on BC and brown carbon is not being considered as another potential explanation.

**FIGURES**

Page 28, Figure 1 – A key should be provided for the wind speed rage shown including units.

Page 30, Figure 4 - I strongly encourage the authors to contact the authors of the previous work so that the real data can be shown in this figure instead of circled approximations.

**REFERENCES**

Suggest adding a space between each citation. Also check that all references are complete, include the journal name or publisher, and that acronyms are spelled out. Specific examples are referenced in the comments above.

Technical Corrections

Page 2, Line 60 – Change "cloud" to "clouds".

Page 4, Line 101 – Forgot degrees symbol before "S"

Page 4, Line 102 – Remove "be" from "The smoke is then be advected...".
**Page 4, Line 110 – Change "was" to "were"**

Page 6, Line 192 – Change "SMPS and PACSP" to "SMPS and PCASP"

---

## Referee Comment (RC2) · Steven Howell (Referee) · 4 Jun 2020

**Review of Wu et al., "Vertical variability of the properties of highly aged biomass burning aerosol transported over the southeast Atlantic during CLARIFY- 2017"**

Steven Howell

**1  Overall**

This is a well-done study of aerosol in an undersampled region. There are numerous interesting findings. It is a valuable addition to the literature. It will undoubtedly be useful for people modeling the effect of very aged BB aerosol on climate.

I have a bunch of minor issues that should be addressed, but nothing that requires wholesale rewriting. There were a couple of points where I would have appreciated more thorough investigation, including the possibility of sampling artifacts, further examination of rBC to OA ratios, and the thermodynamics of $NH_4NO_3$.

**2  Two related issues with nitrates**

The manuscript concludes that most of the nitrate in the aerosol is inorganic. That may well be true, but the evidence offered is a bit slim, as it relies on the relative fractions of ions at $m/z = 30$ and $m/z = 46$. This analysis depends on either a lack of interference at $m/z = 30$ and $m/z = 46$ or on the frag table being correct. The former is a bit shaky, since $CH_2O$ and $C_2H_6$ are at $m/z = 30$ and $CH_2O_2$ and $C_2H_6O$ are at $m/z = 46$. Given the highly oxidized aerosol, $CH_2O$ is likely to be plentiful. The frag table is of course designed for general ambient aerosol and can have striking problems with unusual composition. It is curious that the standard frag table (as of Squirrel 1.62G) assumes there is nothing but $NO_2$ at $m/z = 46$. At $m/z = 30$ there is an indirect dependence on getting the $m/z = 28$ peak right and an assumption that the organic fraction is 2.2%.

(To be fair, I did take a quick look at an arbitrary bit of AMS data from OR-ACLES, where it appeared that $C_2H_6$, $CH_2O_2$ and $C_2H_6O$ were negligible. $CH_2O$ is too close to NO to identify, but the peak was so narrow that significant $CH_2O$ is unlikely. Curiously, there was a bit of $CH_4N$, but not enough to skew the $NO / NO_2$ ratio.) The upshot is that I'd like you to mention that there is some uncertainty in the $m/z30/46$ ratio due to the possibility of organic interference (in addition to the uncertainty in the $m/z30/46$ ratio of organic nitrates.)

The phrase "concentrations of nitrate, ammonium and sulfate ... were governed by the thermodynamic equilibria between their relative mole fractions rather than acidity" isn't clear. The acidity is due to their relative mole fractions. You seem to be saying that concentrations of $SO_4^=$, $NO_3^- + HNO_3$, and $NH_4^+ + NH_3$ are essentially constant and the only change with altitude is that the particulate $NH_4NO_3$ rises as temperature drops and RH rises. Another possibility might be that $NH_3$ is limiting and is more plentiful at higher altitudes.

But the thermodynamics of the $NH_3 + HNO_3 \longleftrightarrow NH_4NO_3$ system have been well known for decades; there is no need to simply wave your hands and say your data are consistent with the thermodynamics when you could do a pretty simple test, going back to Stelson and Seinfeld (1982) or just use the online E-AIM model 4 which uses the work of Friese and Ebel (2010).

Such modeling is particularly relevant because sampling semivolatile aerosols through an aircraft inlet is prone to artifacts. Sampling from aircraft exposes aerosols to a very rapid rise in temperature (due to ram heating as the air is accelerated to aircraft speed) and to extremely intense turbulence. Then there are a few seconds at the temperature of the aircraft interior before entering the instrument. So when sampled, the equilibrium of the $NH_{3v} + HNO_{3v} \leftrightarrow NH_4NO_{3p}$ system will be quite far from ambient. The question is how the time scales for chemical adjustment compare with the transit time within the inlet system. I'm not really sure how to answer. Seinfeld and Pandis (2006) have a section on timescales in the $NH_3v + HNO_3v \leftrightarrow NH_4NO_3$ and conclude that its on the order of minutes in typical situations. Adjustments within particles take microseconds to milliseconds; the limiting factor is the diffusion rate from the aerosol surface to air far from particles. So perhaps aircraft sampling isn't a problem at all, but inlet turbulence is not part of the Seinfeld and Pandis (2006) calculation and would drastically change the effective diffusion rates.

**3 Detailed comments**

**Lines 21–22** The statement "thickly coated black carbon" doesn't belong in the abstract. It sounds like something you'll present in the paper, but you never

actually present any data about coating thickness, just referring to a paper in prep.

**Line 60** "semi-permanent" misspelled

**Line 77** "obscured" isn't exactly right–the aerosol plume is quite visible to the satellites, but the cloud deck makes the intervening aerosol difficult to interpret quantitatively.

**Line 89** What do you mean by "close"? This can easily be quantified, with a statement like "limited data beyond 1000 km from the coast" or something such.

**Line130** "across a complete range of ion mass-charge $(m/z)$ ratios" is meaningless. Did the AMS go to $m/z = 5000$? Just say what masses were scanned. You should also mention the software used for analysis.

**Line 155** A refractive index of $1.54 - 0.027i$ does not appear in the cited Peers et al. (2019) paper. They concluded that the best value was $1.51 - 0.029i$. Something is in error here.

**Line 158–160** The description of SMPS operation is confused. You make reference to Zhou et al., a Ph.D. thesis, so there's no "et al.", and it has little to do with airborne sampling. More plausibly, you used Jingchuan Zhou's inversion scheme and scanned voltages over a 1 minute period. What standards were used to determine whether conditions were constant enough? PCASP concentrations varying no more than 10% or something such?

**Line 167** Impactors deal with aerodynamic diameter. To remove particles $> 1\,\mu\text{m}$ geometric diameter, you must be assuming a particle density. What is it?

**Line 183** I suspect the Lance et al. (2010) isn't what you meant here. Lance (2012) recommend physical changes to the CDP as well as calibration and operation procedures. The modifications to the optics she recommends are pretty standard now, so I suspect your CDP used them, but you ought to mention that.

**Line 183 (and throughout the paper)** The units $\text{g\,m}^{-3}$ and $\text{g\,\mu m}^{-3}$ are ambiguous, since sometimes they refer to cubic meters in the air and sometimes to standard cubic meters (the concentration if the air were at a standard temperature and pressure.) Here I expect you're using actual volume, but usual

practice for AMS measurements is to use standard volume. Please be explicit about what units you use and what definition of "standard" you use.

**Line 210–211** I expect that aerosols were detected at all altitudes in all 3 periods. Do you mean that pollution aerosols were present in both BL and FT during period 3?

**Line 228, 253** $NH_{4\,predicted}^+$ is not a real quantity and is not discussed in Zhang et al. (2007). That paper does talk about $NH_{4\,neu}^+$, which is the $NH_4^+$ concentration if all acids in the aerosol were neutralized. Calling it "predicted" suggests that there is some reason to believe that full neutralization should be the case, which is not at all true. if $[NH_4^+] = NH_{4\,neu}^+$, then it is likely that an excess of $NH_3$ vapor is present. $[NH_4^+] > NH_{4\,neu}^+$ says that there is either a measurement error or that there are un-measured acid species present (presumably organic).

**Line 230** Nitrate formation is not suppressed by un-neutralized $SO_{4=}$, it simply remains as gas phase $HNO_3$.

**Line 247–249** You speculate that lower fraction of OA in aged BB is due to evaporation or oxidation of OA during aging or is due to formation of secondary inorganics. But since you have BC and CO, both of which should be largely conserved, you have the data confirm or rule out the inorganics. You should take a stab at it.

**Line 255** Or the amount of $NO_3$ was simply limited by the available $NH_3$. If it's really thermodynamics of the $HNO_3$, $NH_3$, $NH_4NO_3$, and $H_2O$ system, it ought to be straightforward to do a simple plausibility test with model 4 of E-AIM (Friese and Ebel, 2010) (`http://www.aim.env.uea.ac.uk/aim/model4/mod4t.php`).

**Line 274** "assume" is an inappropriate word; those papers argue that there's evidence that $f60$ lasts longer in the FT.

**Line 278** This paper does not actually provide evidence that $f60$ is lost by oxidation during transport. You don't present any information about what $f60$ is at the source. I have no doubt that $f60$ was higher at the source, but you're making a claim without evidence here. Indeed, one of the lessons of Jolleys et al. (2015) is that $f60$ is highly dependent on fuel and burning conditions. The fairly complete combustion indicated by high MCE is likely to mean that the carbohydrates that make up the $m/z = 60$ signal were largely oxidized.

**Line 286–287** It should be noted that your $f44$ is at the extreme of the data in Aiken et al. (2008) and from very different sources, so the linear fit and error estimates may not be terribly accurate.

**Line 289–290** This is circular reasoning. High $f44$ is the reason you know you have high O/C and OM/OC.

**Line 297** I'm a bit amused that you use the American spelling of "sulfate" in a European journal, then use the English spelling for "smouldering".

**Line 299-300** The text here does not match the description in the supplement or figure 5. The latter two say that BC:CO and OA:CO in the FT were determined from the slope of the relationships, not by subtracting out the background value. Using the slope is more robust to variations in the background levels, so is a good way to do it.

**Line 300 (and Supplement S1)** I appreciate that you're using ODR fits rather than conventional least squares, but ODRs are quite capable of exhibiting artifacts, particularly when the variables have different units (in this case, $\mu g\,m^{-3}$ and ppmv). There's a nice short discussion in the Wikipedia page. For the good correlations you have, it probably makes little difference, but you could easily recast BC to a mole ratio with air or CO ppmv to $\mu g\,m^{-3}$. Or you could wade through Warton et al. (2006) or something similar.

**Line 315-316** The statement "It is likely that observed $BC/\Delta CO$ values in FT smoke are similar to values at source." is likely true, but it should be acknowledged that CO has a lifetime of about a month in the FT due to reaction with OH. That lifetime is pretty long compared with your transit times and may be longer since there are a lot of other things for OH to react with.

**Line 355** Note that the only absorption instrument in SAFARI-2000 was a single-wavelength PSAP. They had to assume an absorption Ångstrom exponent to get SSA at other wavelengths.

**Lines 397–401** It's little irritating to see hand waving here when it could be modeled pretty easily. You might be right that $NH_4NO_3$ would double given the temperature difference, but it could easily be checked for plausibility. Just ignore the organics; they'll be so much less hydrophilic than the sulfate that they won't be responsible for much water.

**Lines 402–407** More hand waving that could be quantified. Given the additional mass of $NH_4NO_3$, what would be a plausible increase in diameter?

**Line 445**  spray

**Supplement line 24** It might be nice to actually say what the background CO concentrations was in the FT.

**Supplement lines 59–70** This analysis depends on either a lack of interference at $m/z = 30$ and $m/z = 46$ or on the frag table being correct. The former is shaky, since $CH_2O$ and $C_2H_6$ are at $m/z = 30$ and $CH_2O_2$ and $C_2H_6O$ are at $m/z = 46$. Given the highly oxidized aerosol, $CH_2O$ is likely to be plentiful. The frag table is of course designed for general ambient aerosol and can have striking problems with unusual composition. It is particularly troubling that the standard frag table for Squirrel 1.62G assumes there is nothing but $NO_2$ at $m/z = 46$. At $m/z = 30$ there is an indirect dependence on getting the $m/z = 28$ peak right. I haven't used a C-ToF, but that peak is so big on an HR-ToF that it seems to be a bit nonlinear. High CO from the aerosol can also screw that up, but I don't think your OA concentrations were that large.

**References**

Aiken, A., P. DeCarlo, J. Kroll, D. Worsnop, J. Huffman, K. Docherty, I. Ulbrich, C. Mohr, J. Kimmel, D. Sueper, et al. (2008). "O/C and OM/OC ratios of primary, secondary, and ambient organic aerosols with high-resolution time-of-flight aerosol mass spectrometry". *Environmental science & technology* 42.12, pp. 4478–4485. ISSN: 0013-936X. DOI: 10.1021/es703009q.

Friese, E. and A. Ebel (Nov. 2010). "Temperature Dependent Thermodynamic Model of the System $H^+$–$NH_4^+$–$Na^+$–$SO_4^{2-}$–$NO_3^-$–$Cl^-$–$H_2O$". *The Journal of Physical Chemistry A* 114.43, pp. 11595–11631. DOI: 10.1021/jp101041j. URL: https://doi.org/10.1021/jp101041j.

Jolleys, M. D., H. Coe, G. McFiggans, J. W. Taylor, S. J. O'Shea, M. Le Breton, S. J.-B. Bauguitte, S. Moller, P. Di Carlo, E. Aruffo, P. I. Palmer, J. D. Lee, C. J. Percival, and M. W. Gallagher (2015). "Properties and evolution of biomass burning organic aerosol from Canadian boreal forest fires". *Atmospheric Chemistry and Physics* 15.6, pp. 3077–3095. DOI: 10.5194/acp-15-3077-2015. URL: https://www.atmos-chem-phys.net/15/3077/2015/.

Lance, S. (2012). "Coincidence Errors in a Cloud Droplet Probe (CDP) and a Cloud and Aerosol Spectrometer (CAS), and the Improved Performance of a Modified CDP". *Journal of Atmospheric and Oceanic Technology* 29.10, pp. 1532–1541. DOI: `10.1175/JTECH-D-11-00208.1`. eprint: `https://doi.org/10.1175/JTECH-D-11-00208.1`. URL: `https://doi.org/10.1175/JTECH-D-11-00208.1`.

Seinfeld, J. H. and S. N. Pandis (2006). *Atmospheric Chemistry and Physics - From Air Pollution to Climate Change (2nd Edition)*. John Wiley & Sons. ISBN: 978-0-471-72018-8.

Stelson, A. W. and J. H. Seinfeld (1982). "Relative humidity and temperature dependence of the ammonium nitrate dissociation constant". *Atmospheric Environment* 16.5, pp. 983–992. ISSN: 0004-6981. DOI: `https://doi.org/10.1016/0004-6981(82)90184-6`. URL: `http://www.sciencedirect.com/science/article/pii/0004698182901846`.

Warton, D. I., I. J. Wright, D. S. Falster, and M. Westoby (2006). "Bivariate line-fitting methods for allometry". *Biological Reviews* 81.2, pp. 259–291. DOI: `10.1017/S1464793106007007`. eprint: `https://onlinelibrary.wiley.com/doi/pdf/10.1017/S1464793106007007`. URL: `https://onlinelibrary.wiley.com/doi/abs/10.1017/S1464793106007007`.

Zhang, Q., J. L. Jimenez, D. R. Worsnop, and M. Canagaratna (May 2007). "A Case Study of Urban Particle Acidity and Its Influence on Secondary Organic Aerosol". *Environmental Science & Technology* 41.9, pp. 3213–3219. DOI: `10.1021/es061812j`. URL: `https://doi.org/10.1021/es061812j`.

---

## Author Comment (AC1) · 10 Jul 2020

**Response to reviewers**

**Firstly, we would like to thank both referees for their important comments, we have addressed all the comments below. The original comments are in black, our replies are in blue and the changes in original manuscript are in red.**

**Besides the corrections based on referees' comments, we also updated the SSA (single scattering albedos) based on the updated PAS (Photo-Acoustic Spectroscopy) absorption data. The updated PAS data were determined from a new calibration protocol that provides more accurate aerosol absorption coefficients. We also calculated the SSA at 550 nm and added related values in the manuscript.**

**Referee #1**

**General Comments:** The authors present vertical information on long-range transport biomass burning (BB) aerosol from Southeastern Africa as sampled from aircraft measurements during CLARIFY-2017. The paper is focused on aerosol optical properties, chemical composition, size distributions, and emission factors. The work is thorough and of great interest in understanding how BB aerosol ages post emission within the boundary layer and free troposphere, especially the direct vertical profile measurements of single scattering albedo and chemical composition. This area of the globe is lacking detailed measurements such as the ones the authors present here, which are the first of their kind and done in a well-organized format. The information on the vertical structure will be very informative for understanding BB ageing and aerosol-cloud interactions for this remote region of the globe to reduce the climate model uncertainties that are very high in this region. I have some comments and suggestions that are focused on improving the impact as well as some technical comments that should be addressed before final publication within the ACP Special Issue: New observations and related modelling studies of the aerosol-cloud-climate system in the Southeast Atlantic and southern Africa regions (ACP/AMT inter-journal SI). Overall, I support this work for publication in ACP.

**Detailed Comments:**
**1 – Page 1, Line 27** – Reporting the lowest SSA values in the FT and the location at 2 km is very noteworthy since as the authors mention, this means the BB aerosol in the region is more absorbing that what is currently used in climate models. Would it be possible to report an average column weighted SSA as well? This would be useful for comparison with satellite data, other passive ground sampling and modeling efforts. If so, this could also be reported for the BL and FT separately as well in the following sentences. If not, then the ranges should be included for the BL and FT.
**2 – Page 1, Line 30** – Recommend stating whether the SSA is highest in the BL or high in the FT since the earlier sentences say the SSA increases in the FT with altitude so it is not clear in the abstract where the highest SSA was observed.
We have rephrased the abstract and added the recommended details.

*The column weighted dry SSAs during CLARIFY were observed to be 0.85, 0.84 and 0.83 at 405, 550 and 658 nm respectively.* We also found significant vertical variation in the dry SSA, as a function of relative chemical composition and size. The lowest

SSA in the column was generally in the low FT layer around 2000 m altitude (averages: 0.82, 0.81 and 0.79 at 405, 550 and 658 nm). This finding is important since it means that BB aerosols across the east Atlantic region are more absorbing than currently represented in climate models, implying that the radiative forcing from BB may be more strongly positive than previously thought. Furthermore, in the FT, average SSAs at 405, 550 and 658 nm increased to 0.87, 0.86 and 0.85 with altitude up to 5 km. This was associated with an enhanced inorganic nitrate mass fraction and aerosol size, likely resulting from increased partitioning of ammonium nitrate to the existing particles at higher altitude with lower temperature and higher relative humidity. After entrainment into the BL, aerosols were generally smaller in dry size than in the FT and had a larger fraction of scattering material with resultant higher average dry SSA, mostly due to marine emissions and aerosol removal by drizzle. *In the BL, the SSA decreased from the surface to the BL top, with the highest SSA in the column observed near the surface.*

**3 – Page 2, Line 54** - What about the other effects on clouds besides CCN formation? Later on, two paragraphs following, the same paper is referenced having calculated direct, indirect and semi-direct effects. It seems disjointed to not mention these effects earlier.

We have added more details about the aerosol effects on clouds. The direct, indirect and semi-direct effects have been mentioned before the cited Gordon et al. (2018).

These smoke layers typically over-lie vast stretches of marine stratocumulus clouds (Adebiyi et al., 2015), where they can exert a warming effect by absorbing both downwelling solar radiation and that scattered upwards from the low-lying clouds (Samset et al., 2013). This *direct* radiative effect is sensitive to the smoke's single-scattering albedo (SSA), which is a function of aerosol composition and size and evolves with particle age (Abel et al., 2005).

The entrained aerosols can affect cloud microphysics by acting as cloud condensation nuclei (CCN), *inducing an indirect radiative effect. In addition, BC immersed in cloud droplets could absorb light and facilitate water evaporation.* BC below clouds could enhance the formation of convection by providing additional heating within the sub-cloud layer. *These effects perturb the temperature structure of the atmosphere and influence the cloud distribution, which are regarded as semi-direct effects* (Koch and Del Genio, 2010; Fan et al., 2018).

**4 – Page 2, Lines 54-55** – Another more recent publication also found significant enhanced convection over the Amazon region due to particles < 50 nm diameter containing BC - Fan et al., Science, 2018 - DOI: 10.1126/science.aan8461).

We have cited the recommended paper.

**5 – Page 2, Line 59** – Suggest changing from "will be important." to "is and will be increasingly important in the future."

Accepted

**6 – Page 3, Line 75 – 77** - What about also the uncertainty in retrievals especially for vertically resolved information? Suggest including this in the manuscript text.

We have added the information "Meanwhile, the vertically resolved retrievals reported by the LASIC campaign using co-located lidar measurements rely on assumptions of the aerosol properties.".

**7 – Page 3, Lines 80-82** - This information is incomplete as written and should be modified. The paper referenced from the LASIC campaign reports the first results of BB aerosol measured during the campaign and focuses on the in situ. The full LASIC campaign included ground-based in situ aerosol measurements as well as column measurements on aerosols and clouds. Retrievals are being processed for column weighted averages as well as to retrieve vertically resolved information. While the vertically resolved retrievals require assumptions and have limitations, there is more data to be analyzed that has yet to be reported in the peer-reviewed literature from LASIC.

We have added more details about the LASIC campaign.

The LASIC (Layered Atlantic Smoke Interactions with Clouds) field campaign was conducted on Ascension Island in the southeast Atlantic, delivering in-situ ground-based aerosol measurements (Zuidema et al., 2018) and column information retrieved from surface-based remote sensing. These measurements were limited to single point or column observations but provide a long and continuous time series. Meanwhile, the vertically resolved retrievals reported by the LASIC campaign using co-located lidar measurements rely on assumptions of the aerosol properties.

**8 - Page 3, Line 89** - Check that all references in the text are listed in the reference section. For example, the Zuidema et al. 2016 reference mentioned here does not appear to be listed in the References Section. - https://journals.ametsoc.org/doi/pdf/10.1175/BAMS-D-15-00082.1

The "Zuidema et al. 2016" is in previous manuscript. We have checked all references again.

**9 - Page 4, Line 129** - Check and improve all reference formats as these 2 Trembath references appear incomplete in the Reference section. It is not clear how someone would find a tech report with no indication of who published it and/or without an associated doi. Also, the abbreviations should be spelled out, for example, SAES is not defined.

We have added more details for Trembath et al. (2012). The Trembath (2012) was deleted.

Trembath, J., Bart, M., and Brooke, J.: FAAM Technical Note: Efficiencies of modified Rosemount housings for sampling aerosol on a fast atmospheric research aircraft, Facility for Airborne Atmospheric Measurements, FAAM, Cranfield, UK, *available at: https://old.faam.ac.uk/index.php/faam-documents/science-instruments/ (last access: 20 May 2020),* 2012.

**10 – Page 5, Line 134-135** – Did the composition dependent CE that was determined based on Middlebrook et al. 2012 for the AMS data here include a mass comparison to the SMPS to validate the CE used?

A mass comparison to the SMPS is not conducted to validate the CE. However, we did compare the (AMS and SP2) with the PCASP. The AMS total mass concentrations were converted to volume concentrations using the densities of 1.27 g cm$^{-3}$ for organics and 1.77 g cm$^{-3}$ for inorganics (Morgan et al., 2010). The SP2 BC mass concentrations were converted to BC volume concentrations using a BC density of 1.8 g cm$^{-3}$ (Liu et al., 2010). Overall, the estimated (AMS and SP2) volume concentrations were 83% to 77% of the estimated PCASP submicron volumes in the BL and FT respectively, which is shown in Figure A. We have added related information in the manuscript.

Comparison of the estimated volumes from the AMS and SP2 with the PCASP was conducted, following the method in Morgan et al. (2010). The total mass concentrations measured from the AMS and SP2 were converted to total volume concentrations,

using densities of 1.27 g cm$^{-3}$ for organics, 1.77 g cm$^{-3}$ for inorganics and 1.8 g cm$^{-3}$ for BC (Morgan et al., 2010, Liu et al., 2010). The submicron volume concentrations from PCASP were estimated using bins with diameter below 1 µm, assuming particles are spherical. The estimated volume from the AMS and SP2 were 83% and 77% of the estimated PCASP volume in the BL and FT respectively. These discrepancies are considered tolerable given the 30–50% uncertainty in PCASP volume estimates (e.g. Moore et al., 2004) and the uncertainty in densities required to convert the AMS mass to volume.

[Figure]

**Figure A: Comparison of calculated volume from the AMS and SP2 with estimated sub-micron volume derived from the PCASP, in the BB-polluted FT and BL separately. Linear regression lines are shown for the FT and BL separately.**

**11 – Page 5, Line 137** – Suggest including references to the early *f*44 and *f*60 work by Cubison et al., ACP, 2011 (already in your references), Ng et al., ACP, 2010 (https://doi.org/10.5194/acp-10-4625-2010), and Ortega et al., 2013 (doi:10.5194/acp13-11551-2013) here where these factors are first mentioned in the text.

We have cited the recommended two papers when *f*44 and *f*60 were first mentioned.

**12 – Page 6, Lines 193 – 194** – How many flights were used out of how many total to make the temperature and specific humidity profiles? This information should be added to the main text and listed in the SI Figure caption.

We have added which flights were used for calculating the vertical profiles of temperature and specific humidity.

The profiles of temperature and specific humidity were derived *from all the flights used in this study (C028-C039, C045-C051)*, as seen in Fig. S1.

**13 – Page 7, Lines 199 – 200** – Can you state in the text how thick the typical inversion layer is between the BL and FT?

We have added the typical thickness of the inversion layer.

The inversion layer sits immediately above the BL and is characterised by a sharp increase in temperature and coincident steep decrease in specific humidity, *typically in a thickness range of 100–400 m.*

**14 – Page 7, Line 225** – The higher SO4 and lower BC are more striking than the 10% change in OA fraction. State here the % changes in these species for the BB-polluted BL Period 1 vs 3 comparison.

We have rephrased to "In the BB-polluted BL (periods 1 and 3), chemical composition ratios showed temporal variations. *The BL in period 1 had ~10% higher average sulfate mass fraction and ~6% lower BC mass fraction than in period 3.*".

**15 – Page 8, Line 227** - With as much as the mass fractions are mentioned in the text, a bar or pie chart with the % mass fractions indicated would be a good visual representation of this comparison that could be added to Figure 3.

We have added pie charts in Fig.3, showing the campaign average values of different chemical composition ratios in the FT and BL separately.

[Figure]

**Figure 3: a) The average vertical distribution of PM1 chemical composition ratios in the BB-polluted FT and BL separately in each period. The width of color bars represents average mass ratio of different species in every 400 m bin. The error bars represent one standard deviation. b) Pie charts showing campaign-average chemical composition ratios in the BB-polluted FT and BL respectively.**

**16 – Page 8, Lines 226 – 227** – State the sulfate fractions in the BL and FT for completeness and a quick comparison in the text so the reader doesn't have to go to the Figures/Tables to get this information.

We have rephrased to "The relative chemical compositions in the BB-polluted BL and FT also showed differences. *Sulfate average mass fractions in the BL were (30 ± 4) % in period 1 and (21 ± 5) % in period 3, which were 2–3 times larger than those in the FT ((11 ± 4) %).*".

**17 – Page 8, Lines 247 – 249** - This is an interesting topic that deserves more attention. A recent publication reviewed lab and field BBA aging of OA and should be included in this discussion is presented by Hodshire et al., ES&T, 2019 (https://doi.org/10.1021/acs.est.9b02588). How common is the observation of secondary inorganic aerosol formation due to aging in ambient and/or laboratory data? Is this unique to BBA?

We have added Hodshire et al., ES&T, 2019.

Previous field studies have observed the inorganic aerosol enhancement of nitrate and sulfate in BB smoke after emission due to the secondary processing of co-emitted $NO_x$ and $SO_2$ (i.e. Yokelson et al., 2009; Pratt et al., 2011; Akagi et al., 2012).

This is not necessarily unique to BB emissions. Inorganic (sulfate and nitrate) species from engine-related emissions were also observed to make an increasing contribution to the total particle mass with age (i.e. Lack et al., 2011)

**18 – Page 9, Line 273** – Consider adding Ortega et al., ACP, 2013 (doi:10.5194/acp13-11551-2013) here in addition to Cubison et al., ACP, 2011.

We have cited the recommended paper.

**19 – Page 9, Line 275** – Is the ∼5 days quoted the timeline of the less aged or more aged BBA? It is not clear in the way the sentence is currently written. Also, how does this relate to the earlier statement that mentions f60 is only a good tracer for BBA with aging timescales < 1 day?

Jolleys et al. (2015) sampled BBA from boreal forest fires at different ages. They observed higher $f60$ ($0.012 \pm 0.005$) in aged smoke that had been transported ∼5 days in the FT than that was reported in near field region at low levels. "The ∼5 days" represented the more aged BBA that contained the higher $f60$. The $f60$ was shown to act as a longer-lived tracer for BB emissions when it is transported to the FT (Jolleys et al., 2015), compared with a previous study which presented that $f60$ decayed to near background value (0.003) for ageing timescales of one day (Cubison et al., 2011). We discussed this, to show that $f60$ would eventually decay to near the background value in the far-field after >7 days transport in this study, although the lofting of BB smoke into the FT may lead to a slower loss of levoglucosan-like species.

We have rephrased this part.

During CLARIFY, average $f60$ were calculated to be $(0.6 \pm 0.3)$ % and $(0.5 \pm 0.2)$ % in the BB-polluted FT and BL respectively. Previous field studies have sampled BBA from flaming fires at Lake McKay (Cubison et al., 2011) and Amazonia fires (Morgan et al., 2020). These previous studies observed much higher $f60$ at source and in the near-source region than that observed in this study. Substantial oxidation and loss of levoglucosan-like species has occurred in the CLARIFY region after >7 days transport. Cubison et al. (2011) observed that $f60$ decayed to near background level (0.3%, in air masses without BB influence) during 1-day transport. The $f60$ is currently thought to be a robust BB tracer for ageing timescales within 1 day from emission (Cubison et al., 2011; Ortega et al., 2013). However, Jolleys et al. (2015) reported an average $f60$ of 1.2% in aged BB smoke that had been transported ∼5 days in the FT after emission from boreal forest fires, well above the 0.3% background level, suggesting that the lofting of BB smoke into the FT may lead to the retention of levoglucosan-like species. The low values presented in this paper indicates $f60$ in the far-field region eventually decayed to near background levels even when the smoke was transported into the FT.

**20 – Page 9, Line 286** – An approximation of the average carbon oxidation state can also be presented using O/C and H/C and Equation 2 in Kroll et al., Nature Chemistry, 2011 - https://www.nature.com/articles/nchem.948

Thank you for the suggestion. We have added the calculation of hydrogen to carbon (H/C) ratio (Ng et al., 2011) and the average carbon oxidation state (OSc) (Kroll et al., 2011). We also plotted the Van Krevelen diagram (H/C vs. O/C) following the method in Ng et al. (2011). We have rephrased the paragraph below:

We also calculated the elemental composition ratios of oxygen to carbon (O/C) and hydrogen to carbon (H/C) based on the estimates proposed by Aiken et al. (2008) and Ng et al. (2011). It should be noted that $f44$ in this study is at the top end of the $f44$ range reported by Aiken et al. (2008) and $f43$ is at the bottom end of the $f43$ range reported by Ng et al. (2011), and the aerosols were sampled from different fire sources in these studies, thus the O/C and H/C may have larger uncertainty than the

reported error of 9% (Aiken et al., 2008) and 10% (Ng et al., 2011). The average carbon oxidation state (OSc) was estimated using O/C and H/C (Kroll et al., 2011). Fig. 4b shows the Van Krevelen diagram (H/C vs. O/C), with average values in each flight and the boundaries of OSc in the BL and FT respectively, following the method in Ng et al. (2011). The elemental composition ratios and OSc were within the observed values of low-volatile oxygenated OA (LV-OOA) (Kroll et al., 2011; Ng et al., 2011).

**21 – Page 9, Line 291** – The aging and oxidation of organic aerosol in the atmosphere towards the formation of LV-OOA and high f44 should also include reference to the work done by Jimenez et al., 2010 - https://science-sciencemagorg.lanl.idm.oclc.org/content/326/5959/1525.
We have cited the recommended paper.

**22 – Page 10, Lines 312 – 314** – How is it known that all "particles observed in the FT have not encountered cloud" during CLARIFY? Please explain, elaborate or reference previous work to substantiate the claim.
We have rephrased to "*Particles are unlikely to have been subject to significant wet removal after being lofted into the FT, due to the low water contents and low probability of encountering clouds in the FT over the southeast Atlantic. Hence the FT aerosols are likely to be long-lived.*".

**23 – Page 10, Lines 320 – 323** – The assumption seems to be that the BBA in the BL is the result of mixing between BBA that was lofted and transported in the FT via downdrafts into the BL. Is there relevant work that can be cited to support these claims for this region?
Back-trajectories released from BL at Ascension Island location were analysed in Zuidema et al. (2018) and Haywood et al. (2020) and showed that the observed BL BBA originated from BB plumes over south Africa. Climate model simulations made by Gordon et al. (2018) also showed that the BB smoke from Africa could mix into the BL. We have added this into the manuscript.
*Back-trajectories initiated from Ascension Island (Zuidema et al., 2018; Haywood et al., 2020) and climate model simulations made by Gordon et al. (2018) indicated that the BB smoke from south African fires had entrained into the marine BL over the southeast Atlantic. BC/ΔCO and OA/ΔCO ratios in the BL were generally lower than that in the FT, clearly seen throughout period 3,* indicating that a fraction of particles may be removed by cloud activation or scavenging and subsequent precipitation after FT aerosols mix into the BL.

**24 – Page 11, Line 332** – Suggest changing "indicates that new particle formation was occurring. . ." to "indicates that new particle formation and growth was likely occurring".
Accepted

**25 - Page 14, Line 420** – Why are marine emissions and cloud processing considered one factor?
Dimethyl sulfide (DMS), from oceanic biogenic emission is an important source of global sulfate aerosol (Perraud et al., 2015). DMS can be oxidized into aerosol precursors, i.e. $SO_2$ and methane sulfonic acid (MSA). The further $SO_2$ reaction with OH

radical in the gas phase produce $H_2SO_{4(g)}$ molecules, which would either form new particles or condense on to existing aerosol surfaces. *Removal of SO₂ due to in-cloud processing also provides a sulfate source.*

For these reasons, we state "The first factor is marine emissions and cloud processing in the BL" in the previous manuscript. We have removed "and cloud processing" to avoid misleading the reader.

**26 – Page 14, Lines 420 – 421** – Does "removal processes" include wet and dry deposition?

We refer to wet deposition, as we explained in the manuscript "The wet removal events usually occur via aerosol activation to form cloud droplets during in-cloud processing and subsequent removal of those droplets by precipitation, which would also facilitate below-cloud aerosol scavenging".

**27 – Page 14, Line 421** – Same as earlier comment – suggestion to restate to include "new particle formation and growth".

Accepted

**28 – Page 14, Line 433** – Consider adding Heald et al., GRL, 2010 when referencing carboxylic acid content in addition to Duplissy et al., 2011). - https://agupubs.onlinelibrary.wiley.com/doi/full/10.1029/2010GL042737

We have cited the recommended paper.

**29 – Page 15, Lines 475 – 477** – State why enhanced MAC's are being attributed to coatings on BC and brown carbon is not being considered as another potential explanation.

*The absorption angstrom exponent (AAE) of CLARIFY BBA was reported to be close to 1 (Taylor et al., 2020). It is assumed that an AAE over 1 indicates absorption from particles like brown carbon (BrC) or dust which have higher AAEs than BC (Lack and Langridge, 2013). There was only a minor BrC contribution to the total aerosol absorption in CLARIFY region (Taylor et al., 2020).* The enhanced absorption *is therefore likely to be mostly* due to the observed thick coatings on BC (Taylor et al., 2020), causing a lensing effect and additional absorption of sunlight (Lack et al., 2009).

**FIGURES**

Page 28, Figure 1 – A key should be provided for the wind speed rage shown including units.

[Figure]

**Figure 1: (a) The integrated spatial distribution of MODIS-detected fire counts in August 2017, coupled with flight tracks (without transit flights) during CLARIFY-2017 aircraft campaign (16th August-7th September). (b) Average wind speed and direction at 925hPa (left) and 700hPa (right) in ERA-Interim re-analysis for August 2017. The wind speed was shown in grey scale bars.**

Page 30, Figure 4 – I strongly encourage the authors to contact the authors of the previous work so that the real data can be shown in this figure instead of circled approximations.

[Figure]

**Figure 4: a) The fractional signals *f*44 vs *f*60 of aerosols sampled in this and other studies. Blue and black markers represent the average of FT and BL BBA layers, respectively, for each flight. The vertical dashed grey line indicates the background of *f*60 (0.3%) under non-BB conditions, as recommended by Cubison et al. (2011). b) The average and standard deviation of H/C vs. O/C in each flight and the boundaries of OSc in the BL (black) and FT (blue) respectively.**

**REFERENCES** Suggest adding a space between each citation. Also check that all references are complete, include the journal name or publisher, and that acronyms are spelled out. Specific examples are referenced in the comments above. **Technical Corrections**

Page 2, Line 60 – Change "cloud" to "clouds".

Page 4, Line 101 – Forgot degrees symbol before "S"

Page 4, Line 102 – Remove "be" from "The smoke is then be advected. . .".

Page 4, Line 110 – Change "was" to "were" Page 6, Line 192 – Change "SMPS and PACSP" to "SMPS and PCASP"

We have checked references and corrected these technical issues.

**Referee #2**

**Overall**: This is a well-done study of aerosol in an undersampled region. There are numerous interesting findings. It is a valuable addition to the literature. It will undoubtedly be useful for people modeling the effect of very aged BB aerosol on climate. I have a bunch of minor issues that should be addressed, but nothing that requires wholesale rewriting. There were a couple of points where I would have appreciated more thorough investigation, including the possibility of sampling artifacts, further examination of rBC to OA ratios, and the thermodynamics of $NH_4NO_3$.

**Two related issues with nitrates:**

**1.** The manuscript concludes that most of the nitrate in the aerosol is inorganic. That may well be true, but the evidence offered is a bit slim, as it relies on the relative fractions of ions at m/z = 30 and m/z = 46. This analysis depends on either a lack of interference at m/z = 30 and m/z = 46 or on the frag table being correct. The former is a bit shaky, since $CH_2O$ and $C_2H_6$ are at m/z = 30 and $CH_2O_2$ and $C_2H_6O$ are at m/z = 46. Given the highly oxidized aerosol, $CH_2O$ is likely to be plentiful. The frag table is of course designed for general ambient aerosol and can have striking problems with unusual composition. It is curious that the standard frag table (as of Squirrel 1.62G) assumes there is nothing but $NO_2$ at m/z = 46. At m/z = 30 there is an indirect dependence on getting the m/z = 28 peak right and an assumption that the organic fraction is 2.2%. 1 (To be fair, I did take a quick look at an arbitrary bit of AMS data from ORACLES, where it appeared that $C_2H_6$, $CH_2O_2$ and $C_2H_6O$ were negligible. $CH_2O$ is too close to NO to identify, but the peak was so narrow that significant $CH_2O$ is unlikely. Curiously, there was a bit of $CH_4N$, but not enough to skew the NO / $NO_2$ ratio. The upshot is that I'd like you to mention that there is some uncertainty in the m/z30/46 ratio due to the possibility of organic interference (in addition to the uncertainty in the m/z30/46 ratio of organic nitrates.)

Thank you for the suggestion of adding discussion of uncertainty in the m/z 30/46 ratio due to the possibility of organic interference in the C-ToF AMS. We have added this in the Supplementary S2.

Nitrate is detected in the AMS using peaks at m/z = 30 and 46 (Allan et al., 2003), representing the ions $NO^+$ and $NO_2^+$ respectively. The AMS can detect nitrate species including inorganics like $NH_4NO_3$, $NaNO_3$ and $KNO_3$, as well as organic

nitrates. When sampling different nitrate species, the ratio of these two peaks is determined by the heater temperature and the volatility of nitrate species (Drewnick et al., 2015). Higher ratios were observed for less volatile nitrates, e.g. 28 for $KNO_3$ and 29.2 for $NaNO_3$, compared to $NH_4NO_3$, since they decompose further before ionization (Alfarra et al., 2004; Drewnick et al., 2015). Rollins et al. (2010) measured ratios of 1.51 – 5.10 for various organic nitrates. During CLARIFY, the m/z 30 to m/z 46 ratios ranged from 1 to 1.4, from the AMS calibration using mono-disperse $NH_4NO_3$ particles. The vertical profile of ambient m/z 30 to m/z 46 ratios in CLARIFY FT was shown in Fig. S3. *With the C-ToF AMS used in this study, the interference of some ions from organics cannot be separated at these two peaks, such as $CH_2O^+$, $CH_4N^+$ and $C_2H_6^+$ at m/z = 30 and $CH_2O_2^+$ and $C_2H_6O^+$ at m/z = 46, which would add uncertainties in the ambient m/z 30 to m/z 46 ratios for nitrate.* However, given the small difference between ambient and calibration values, there is likely a low potential interference from large amounts of organic nitrates, and most of observed nitrates would be $NH_4NO_3$. Furthermore, the linear fitted $NH_{4mea}^+/NH_{4neu}^+$ ratios (in which $NH_{4mea}^+$ is the measured ammonium concentration, $NH_{4neu}^+$ is the calculated ammonium concentration if all acids in the aerosol were neutralized) of FT pollutants in period 2 and 3 were $(1.06 \pm 0.01)$ and $(1.05 \pm 0.02)$ respectively (Zhang et al., 2007). The ammonium in the FT was sufficient to nearly fully neutralize the aerosol, which further supports our inference that the nitrate measured in the FT was mostly $NH_4NO_3$.

**2.** The phrase "concentrations of nitrate, ammonium and sulfate ... were governed by the thermodynamic equilibria between their relative mole fractions rather than acidity" isn't clear. The acidity is due to their relative mole fractions. You seem to be saying that concentrations of $SO_4^=$, $NO_3^- + HNO_3$, and $NH_4^+ + NH_3$ are essentially constant and the only change with altitude is that the particulate $NH_4NO_3$ rises as temperature drops and RH rises. Another possibility might be that $NH_3$ is limiting and is more plentiful at higher altitudes. But the thermodynamics of the $NH_3 + HNO_3 \longleftrightarrow NH_4NO_3$ system have been well known for decades; there is no need to simply wave your hands and say your data are consistent with the thermodynamics when you could do a pretty simple test, going back to Stelson and Seinfeld (1982) or just use the online E-AIM model 4 which uses the work of Friese and Ebel (2010).

Such modeling is particularly relevant because sampling semivolatile aerosols through an aircraft inlet is prone to artifacts. Sampling from aircraft exposes aerosols to a very rapid rise in temperature (due to ram heating as the air is accelerated to aircraft speed) and to extremely intense turbulence. Then there are a few seconds at the temperature of the aircraft interior before entering the instrument. So when sampled, the equilibrium of the $NH_{3v} + HNO_{3v} \leftrightarrow NH_4NO_{3p}$ system will be quite far from ambient. The question is how the time scales for chemical adjustment compare with the transit time within the inlet system. I'm not really sure how to answer. Seinfeld and Pandis (2006) have a section on timescales in the $NH_{3v} + HNO_{3v} \leftrightarrow NH_4NO_3$ and conclude that its on the order of minutes in typical situations. Adjustments within particles take microseconds to milliseconds; the limiting factor is the diffusion rate from the aerosol surface to air far from particles. So perhaps aircraft sampling isn't a problem at all, but inlet turbulence is not part of the Seinfeld and Pandis (2006) calculation and would drastically change the effective diffusion rates.

Thank you for these suggestions:

1) We have re-phrased the "concentrations of nitrate, ammonium and sulfate ... were governed by the thermodynamic equilibria between their relative mole fractions rather than acidity". What we describe here, is that the sums of the charges

of nitrate, ammonium and sulfate in the system reach balance. The discussion of thermodynamic process is all put into Section 4.1.1.

In the BB-polluted FT, the linear fitted C-ToF AMS $NH_{4mea}^+/NH_{4neu}^+$ ratios of aerosols in period 2 and 3 were $(1.06 \pm 0.01)$ and $(1.05 \pm 0.02)$ respectively, indicating that sulfate was fully neutralized and nitrate aerosol was formed with the excess ammonia. *Therefore, the amounts of measured nitrate, ammonium and sulfate reached ion balance in the FT. When the observed nitrate mass fraction increased with altitude (mean values ranged from 4% to 13% in period 2 and from 6% to 11% in period 3 respectively), the sulfate mass fraction was relatively constant, the ammonium mass fraction consistently increased with altitude.*

2) Thank you for the suggestion of performing model tests of the thermodynamic processes. We performed a case study using the online E-AIM model 4 (Friese and Ebel, 2010).

During some flights, individual layers were well-mixed, indicated by a constant potential temperature throughout their depth. In these smoke layers, increasing mass concentrations of nitrate and ammonium with increasing altitude were observed, while other species were relatively invariant with altitude. *An example of well-mixed smoke layer from flight C036 is shown in Fig. 10a, b. We conducted a simulation of the chemical thermodynamics in this example smoke layer, using a temperature dependent thermodynamic model described in Friese and Ebel (2010). The inputs of ambient conditions (temperature and water content) and inorganic compositions (sulfate, nitrate and ammonium) were set to measured values in flight C036, the ammonia value was assumed from previous savanna studies in Andreae (2019). With the same initial compositions, the nitrate and ammonium concentrations were simulated over a measured temperature range (at different altitudes) from 281 to 269K. The modelled nitrate and ammonium showed an increasing trend with height, in a similar way to that of the measurements (see Fig. 10c), suggesting lower temperatures at higher altitudes would shift the gas-particle partitioning of $HNO_3$-$NH_3$-$NH_4NO_3$ system toward the aerosol phase and significantly increase the amount of $NH_4NO_3$.* The intrusion of BB smoke in the FT during periods 2 and 3 increased specific humidity compared with the cleaner FT in period 1 (see Fig. 9), since the FT smoke tends to coexist with enhanced water vapor as discussed in Adebiyi et al. (2015). *With relatively constant specific humidity in BB smoke over the vertical profile, the measured and simulated RH (Fig. 9 and 10c) were both shown to increase at higher altitudes, consistent with colder temperatures aloft. As RH values reaching 70%, i.e. at the top of the aerosol layers around 5 km*, aerosols are likely to become liquid particles and allow $NH_4NO_3$ to dissolve in the aqueous aerosol phase. In summary, there is a greater chance for nitrate to be present in the aerosol phase in the colder and higher RH atmosphere encountered towards the top of the aerosol layers.

[Figure]

**Figure 10: The vertical distribution (3000 – 5000 m) of a) different chemical composition concentrations and b) potential temperature and specific humidity in flight C036. The lines and shades represent the 25%, median and 75% in every 200m bin. c) The simulated and measured chemical composition and RH at different altitudes with variable temperatures.**

3)   Thank you for the suggestion of adding the discussion of the possibility of sampling artifacts due to semi-volatile species. We have added this into supplementary S2.

When the air is sampled into the aircraft inlet, it undergoes a rapid temperature rise before entering the AMS inlet due to ram heating as the air is accelerated on sampling. $NH_4NO_3$ is a semi-volatile species, and the rapid change in temperature will influence the thermodynamic equilibrium of $HNO_3$-$NH_3$-$NH_4NO_3$ system, causing the evaporation of $NH_4NO_3$. On the BAe-146 Atmospheric Research Aircraft (ARA), the transport time of sampled air between aircraft inlet and the AMS inlet is ~1–2 s. The timescale for aerosol equilibrium between gas and particle phase is expected to be a few minutes or less under typical polluted conditions (Wexler and Seinfeld, 1992). Therefore, the 1–2 s timescale of heating in the aircraft sampling inlet is sufficiently fast that the partitioning of $NH_4NO_3$ is not influenced. The observed vertical variation of nitrate mass fraction should reflect the influence of temperature change under ambient conditions, which we discussed in section 4.1.1 in the manuscript.

**Detailed comments:**

**Lines 21–22** The statement "thickly coated black carbon" doesn't belong in the abstract. It sounds like something you'll present in the paper, but you never actually present any data about coating thickness, just referring to a paper in prep.

We have removed this from the abstract.

**Line 60** "semi-permanent" misspelled

Accepted

**Line 77** "obscured" isn't exactly right–the aerosol plume is quite visible to the satellites, but the cloud deck makes the intervening aerosol difficult to interpret quantitatively.

We have re-phrased this part. *Satellite-based observations have been employed in this region, but the ability of satellites to quantify BBA amount and its microphysical and optical properties in the marine BL is limited, since the presence of intervening cloud layers brings significant challenges to retrievals of aerosol properties.*

**Line 89** What do you mean by "close"? This can easily be quantified, with a statement like "limited data beyond 1000 km from the coast" or something such.

We have re-phrased this sentence.

More recently, the NASA ORACLES (ObseRvations of Aerosols above CLouds and their intEractionS) campaigns in September 2016, August to September 2017 and October 2018 extended measurements over the south Atlantic, *mostly sampling westward of the SAFARI region and eastward of 0°E* (Zuidema et al., 2016; Pistone et al., 2019).

**Line130** "across a complete range of ion mass-charge (m/z) ratios" is meaningless. Did the AMS go to m/z = 5000? Just say what masses were scanned. You should also mention the software used for analysis.

We have added the suggested details in the manuscript.

The chemical composition of non-refractory submicron aerosols was measured by a compact time-of-flight aerosol mass spectrometer (C-ToF AMS, Aerodyne Research Inc, Billerica, MA, USA) (Drewnick et al., 2005)*, which provides chemical characterization across a range of ion mass-charge (m/z) ratios from 10 to 500.*
*The AMS data was processed using the standard SQUIRREL (SeQUential Igor data RetRiEvaL, v.1.60N) ToF-AMS software package.*

**Line 155** A refractive index of 1.54−0.027i does not appear in the cited Peers et al. (2019) paper. They concluded that the best value was 1.51−0.029i. Something is in error here.

The Peers et al. (2019) refractive index was subsequently updated and we have used the revised value.

Mie scattering theory was used to determine the bin sizes by assuming particles are spherical, with a refractive index of 1.54 – 0.027i. *The refractive index was obtained by the methods reported by Peers et al. (2019), where the aerosol model best represents the PCASP measurement.*

**Line 158–160** The description of SMPS operation is confused. You make reference to Zhou, a Ph.D. thesis, and it has little to do with airborne sampling. More plausibly, you used Jingchuan Zhou's inversion scheme and scanned voltages over a 1minute period. What standards were used to determine whether conditions were constant enough? PCASP concentrations varying no more than 10% or something such?

We used the inversion scheme in Zhou (2001). We only selected straight and runs, where AMS and SP2 data generally demonstrate that species concentrations varied less than 20%. We have rephrased this part.

*The SMPS data was inverted using the scheme developed by Zhou (2001), based on a ~1 min averaging time only during straight and level runs when AMS and SP2 concentrations generally varied less than 20%.*

**Line 167** Impactors deal with aerodynamic diameter. To remove particles > 1 µm geometric diameter, you must be assuming a particle density. What is it?

An impactor removed particles with aerodynamic diameter > 1.3 µm (Brechtel, custom design) (Davies et al., 2019). For pressures between approximately 1000–700 hPa, the geometric diameter is estimated to be ~ 1µm assuming an average aerosol density of 1.6 g cm$^{-3}$ and a particle shape parameter of 1.0. We have changed to aerodynamic diameter in the manuscript.

**Line 183** I suspect the Lance et al. (2010) isn't what you meant here. Lance (2012) recommend physical changes to the CDP as well as calibration and operation procedures. The modifications to the optics she recommends are pretty standard now, so I suspect your CDP used them, but you ought to mention that.
We also added Lance (2012) in the manuscript.

**Line 183** (and throughout the paper) The units g m$^{-3}$ and g µm$^{-3}$ are ambiguous, since sometimes they refer to cubic meters in the air and sometimes to standard cubic meters (the concentration if the air were at a standard temperature and pressure.) Here I expect you're using actual volume, but usual practice for AMS measurements is to use standard volume. Please be explicit about what units you use and what definition of "standard" you use.
We have clarified in the start of section 2.2., that "All measurements reported here were corrected to standard temperature and pressure (STP, 273.15K and 1013.25 hPa)". So, the units g m$^{-3}$ and g µm$^{-3}$ refer to standard.
We have changed these to g sm$^{-3}$ and µg sm$^{-3}$.

**Line 210–211** I expect that aerosols were detected at all altitudes in all 3 periods. Do you mean that pollution aerosols were present in both BL and FT during period 3?
We have changed to "During period 3 (from 29$^{th}$ Aug to 5$^{th}$ Sep, C045-C051), *the BB pollution* was observed throughout the BL and FT.".

**Line 228, 253** NH$_4^+$$_{predicted}$ is not a real quantity and is not discussed in Zhang et al. (2007). That paper does talk about NH$_4^+$$_{neu}$, which is the NH$_4^+$ concentration if all acids in the aerosol were neutralized. Calling it "predicted" suggests that there is some reason to believe that full neutralization should be the case, which is not at all true. if [NH$_4^+$] = NH$_4^+$$_{neu}$, then it is likely that an excess of NH$_3$ vapor is present. [NH$_4^+$] > NH$_4^+$$_{neu}$ says that there is either a measurement error or that there are un-measured acid species present (presumably organic).
We have changed the NH$_4^+$$_{predicted}$ to NH$_4^+$$_{neu}$ in the manuscript, to be consistent with Zhang et al. (2007).
The linear fitted NH$_{4mea}^+$/NH$_{4neu}^+$ ratios in the BL were $(0.86 \pm 0.01)$ and $(0.99 \pm 0.02)$ for period 1 and 3 respectively, indicating the possible presence of acidic aerosol during the first period *(the NH$_{4mea}^+$ is the measured ammonium concentration from the AMS, the NH$_{4neu}^+$ is the calculated ammunium concentration if all acids in the aerosol were neutralized)* (Zhang et al., 2007). In the BB-polluted FT, *the linear fitted C-ToF AMS NH$_{4mea}^+$/NH$_{4neu}^+$ ratios* of aerosols in period 2 and 3 were $(1.06 \pm 0.01)$ and $(1.05 \pm 0.02)$ respectively, indicating that sulfate was fully neutralized and nitrate aerosol was formed with the excess ammonia.

**Line 230** Nitrate formation is not suppressed by un-neutralized SO$_4^=$, it simply remains as gas phase HNO$_3$.

We have rephrased to "When sulfate is not fully neutralized, *nitrate aerosol formation* is suppressed due to the absence of excess of ammonia.".

**Line 247–249** You speculate that lower fraction of OA in aged BB is due to evaporation or oxidation of OA during aging or is due to formation of secondary inorganics. But since you have BC and CO, both of which should be largely conserved, you have the data confirm or rule out the inorganics. You should take a stab at it.

The measurements during CLARIFY were all made in the far-field region, the sampled BB smoke has undergone many days after emission from source fires. The BC and CO are likely to be largely conserved over the timescales of transport from Africa to CLARIFY region. The enhancement ratios of OA and inorganics over BC or CO could indicate the evolution of components after emission, in the absence of wet removal during transport. However, we only sampled highly aged BB aerosols, it is unlikely to observe chemical evolution process in the far-field region. And, we do not have the source or near-source information of OA/BC (or OA/CO) and inorganics/BC (or inorganics/CO), to compare with the CLARIFY region. It is hard to use the ratios to confirm or rule out the inorganic. Both the ageing loss of OA and the formation of secondary inorganics may contribute to the lower fraction of OA we observed relative to the near field measurements cited in other work. The observations from ORACLES may be helpful to disentangle this since the measurements were taken closer to source. We will focus on this question if we have more source or near-source data in the future.

**Line 255** Or the amount of $NO_3$ was simply limited by the available $NH_3$. If it's really thermodynamics of the $HNO_3$, $NH_3$, $NH_4NO_3$, and $H_2O$ system, it ought to be straightforward to do a simple plausibility test with model 4 of EAIM (Friese and Ebel, 2010) (http://www.aim.env.uea.ac.uk/aim/model4/ mod4t.php).

Thank you for the suggestion. We have rephrased to "the sums of the charges of nitrate, ammonium and sulfate in the system reach balance".

The discussion of thermodynamic process is all put into Section 4.1.1. With regards to the modelling using E-AIM, we performed E-AIM model calculations of the nitrate, sulfate and ammonium partitioning with temperature and humidity, which demonstrate similar trends to those shown in our measurements. Related answer can be found in the reply of "**Two related issues with nitrates: 2, 2)**".

**Line 274** "assume" is an inappropriate word; those papers argue that there's evidence that f60 lasts longer in the FT.
**Line 278** This paper does not actually provide evidence that f60 is lost by oxidation during transport. You don't present any information about what f60 is at the source. I have no doubt that f60 was higher at the source, but you're making a claim without evidence here. Indeed, one of the lessons of Jolleys et al. (2015) is that f60 is highly dependent on fuel and burning conditions. The fairly complete combustion indicated by high MCE is likely to mean that the carbohydrates that make up the m/z = 60 signal were largely oxidized.

Thank you for the suggestion. We have changed "assume" to "suggest" and rephrased the discussion of "longer lasting *f*60 in the FT". We have presented *f*60 from measurements of BB at source and near-source region (Cubison et al., 2011; Morgan et al., 2020), to indicate high *f*60 at source and substantial loss of *f*60 by oxidation during transport in CLARIFY.

During CLARIFY, average $f60$ were calculated to be $(0.6 \pm 0.3)$ % and $(0.5 \pm 0.2)$ % in the BB-polluted FT and BL respectively. Previous field studies have sampled BBA from flaming fires at Lake McKay (Cubison et al., 2011) and Amazonia fires (Morgan et al., 2020). These previous studies observed much higher $f60$ at source and in the near-source region than that observed in this study. Substantial oxidation and loss of levoglucosan-like species has occurred in the CLARIFY region after >7 days transport. Cubison et al. (2011) observed that $f60$ decayed to near background level (0.3%, in air masses without BB influence) during 1-day transport. The $f60$ is currently thought to be a robust BB tracer for ageing timescales within 1 day from emission (Cubison et al., 2011; Ortega et al., 2013). However, Jolleys et al. (2015) reported an average $f60$ of 1.2% in aged BB smoke that had been transported ~5 days in the FT after emission from boreal forest fires, well above the 0.3% background level, suggesting that the lofting of BB smoke into the FT may lead to the retention of levoglucosan-like species. The low values presented in this paper indicates $f60$ in the far-field region eventually decayed to near background levels even when the smoke was transported into the FT.

**Line 286–287** It should be noted that your f44 is at the extreme of the data in Aiken et al. (2008) and from very different sources, so the linear fit and error estimates may not be terribly accurate.
The suggestion has been added in the manuscript.
It should be noted that $f44$ in this study is at the top end of the $f44$ range reported by Aiken et al. (2008) and $f43$ is at the bottom end of the $f43$ range reported by Ng et al. (2011), and the aerosols were sampled from different fire sources in these studies, thus the O/C and H/C may have larger uncertainty than the reported error of 9% (Aiken et al., 2008) and 10% (Ng et al., 2011).

**Line 289–290** This is circular reasoning. High f44 is the reason you know you have high O/C and OM/OC.
Combining another referee's suggestion of adding the hydrogen to carbon (H/C) and Oxidation State in Kroll et al. (2011) and Ng et al. (2011), we have rephrased this part.
In general, ageing increases the oxidation state of OA, associated with increasing $f44$, O/C and OM/OC ratios (Jimenez et al., 2009). These high values consistently reflect the highly oxidized and low volatility nature of BB OA in the CLARIFY region.

**Line 297** I'm a bit amused that you use the American spelling of "sulfate" in a European journal, then use the English spelling for "smouldering".
We use "sulfate" since it's the IUPAC standard name for spelling this chemical element. We have united to the English spelling, except for "sulfate".

**Line 299-300** The text here does not match the description in the supplement or figure 5. The latter two say that BC:CO and OA:CO in the FT were determined from the slope of the relationships, not by subtracting out the background value. Using the slope is more robust to variations in the background levels, so is a good way to do it.
We have rephrased to "The enhancement ratios of BC and OA over CO (BC/$\Delta$CO and OA/$\Delta$CO, μg m$^{-3}$ / μg m$^{-3}$) were *calculated in the FT by the unconstrained linear orthogonal distance regression (ODR) fit* (Yokelson et al., 2013), *and were*

*calculated in the BL* by dividing them by the excess concentration of CO, after background values had been removed (Lefer et al., 1994). The detailed calculation method is listed in Supplementary S1.".

**Line 300 (and Supplement S1)** I appreciate that you're using ODR fits rather than conventional least squares, but ODRs are quite capable of exhibiting artifacts, particularly when the variables have different units (in this case, µg m$^{-3}$ and ppmv). There's a nice short discussion in the Wikipedia page. For the good correlations you have, it probably makes little difference, but you could easily recast BC to a mole ratio with air or CO ppmv to µg m$^{-3}$. Or you could wade through Warton et al. (2006) or something similar.

Thank you for the suggestion. We have united the BC, OA and CO units to (µg m$^{-3}$), and re-did the ODR fit in the FT and also the calculation in the BL. The units of BC/ΔCO and OA/ΔCO values are (µg m$^{-3}$/ µg m$^{-3}$). The values and Fig.5 have been updated in the manuscript.

For example, BC/ΔCO values from *0.005 to 0.023* and OA/ΔCO values from *0.037 to 0.066* were observed for BB source in flaming combustion from previous measurements (May et al., 2014; Pratt et al., 2011). A range of *(0.0014 – 0.0072) for BC/ΔCO and (0.080 – 0.096)* for OA/ΔCO, were reported for BB source in smouldering combustion (Capes et al., 2008; Kondo et al., 2011; May et al., 2014).

The calculated BC/ΔCO and OA/ΔCO ratios in FT and BL smoke for each flight are shown in Fig. 5. In the BB-polluted FT, the ODR fitted BC/ΔCO ratios ranged from *0.0087 to 0.0114* in period 2 and were higher *(0.0103 – 0.0134)* in period 3, while OA/ΔCO values were comparable between the two periods *(period 2: 0.042 – 0.067; period 3: 0.043 – 0.064).* In the BB-polluted BL, the average BC/ΔCO and OA/ΔCO ratios in period 1 *(0.0103 – 0.0111; 0.062 – 0.079)* were higher than in period 3 *(0.006 – 0.0085; 0.024 – 0.041).*

[Figure]

**Figure 5: Top panel: the calculated MCE of CLARIFY FT smoke plumes for each flight. The error bars show the uncertainty. Middle and bottom panels: the calculated BC/ΔCO and OA/ΔCO in FT and BL smoke plumes for each flight, the blue markers and error bars represent the fitted slopes and uncertainty in the FT, the black markers and errors represent the average and standard deviation of calculated ratios in the BL.**

**Line 315-316** The statement "It is likely that observed BC/ΔCO values in FT smoke are similar to values at source." is likely true, but it should be acknowledged that CO has a lifetime of about a month in the FT due to reaction with OH. That lifetime is pretty long compared with your transit times and may be longer since there are a lot of other things for OH to react with.

Thank you for the suggestion, we have added the comment in the manuscript.

Particles are unlikely to have been subject to significant wet removal after being lofted into the FT, due to the low water contents and low probability of encountering clouds in the FT over the southeast Atlantic. Hence the FT aerosols are likely to be long-lived. *It is also acknowledged that CO has a lifetime of ~a month by gas-phase oxidation; this lifetime is much longer than the transport timescales in this study.* Previous studies have observed the transatlantic transport of BB pollutants from Africa to the Amazon Basin, reporting a BC/ΔCO value of 0.0117 in FT transported smoke, which is within the observed range in this study (Baars et al., 2011; Holanda et al., 2020). For CLARIFY, it is likely that BC/ΔCO in FT smoke are similar to values at source.

**Line 355** Note that the only absorption instrument in SAFARI-2000 was a single wavelength PSAP. They had to assume an absorption °Angstrom exponent to get SSA at other wavelengths.

We have added the note in Fig.8, "Note: The PSAP only measured absorption at 567 nm in SAFARI-2000, assumptions about the wavelength dependence of absorption coefficient were made to estimate absorption at 450 and 700 nm, which was then used to calculate the SSA.".

**Lines 397–401** It's little irritating to see hand waving here when it could be modeled pretty easily. You might be right that $NH_4NO_3$ would double given the temperature difference, but it could easily be checked for plausibility. Just ignore the organics; they'll be so much less hydrophilic than the sulfate that they won't be responsible for much water.

Thank you for the suggestion. We have performed a case study using the online E-AIM model 4 (Friese and Ebel, 2010). The inputs of ambient conditions (temperature and water content) and inorganic compositions (sulfate, nitrate and ammonium) were set to measured values from a well-mixed smoke layer in flight C036, the ammonia value was assumed from previous savanna studies in Andreae (2019). This simulation showed a similar trend as the measurements, demonstrating that lower temperatures would shift the gas-particle partitioning system of $HNO_3$-$NH_3$-$NH_4NO_3$ toward the aerosol phase and increase the amount of $NH_4NO_3$. This has been answered in the reply of "**Two related issues with nitrates: 2, 2**)".

**Lines 402–407** More hand waving that could be quantified. Given the additional mass of $NH_4NO_3$, what would be a plausible increase in diameter?

The average aerosol composition fractions in the low FT were observed to be 64%, 14%, 6%, 12% and 4% for OA, BC, ammonium, sulfate and nitrate. The average density of a particle in the low FT was estimated to be 1.356 g cm$^{-3}$ following the Zdanovskii, Stokes and Robinson (ZSR) mixing rule and densities of different species used in Haslett et al. (2019), assuming all particles are internally mixed. Over the vertical profiles from low FT up to 5 km, the mass was observed to have a ~15% increase, due to the additional mass of $NH_4NO_3$. The density of $NH_4NO_3$ is assumed to be 1.725 g cm$^{-3}$ (Haslett et al., 2019).

With the ~15% increase of mass, there is estimated to be a ~4% increase in the aerosol radius. We have added this in the manuscript.

From the low FT up to 5 km, the mass was observed to increase by ~15% due to the additional $NH_4NO_3$. The average aerosol composition fractions in the low FT were observed to be 64%, 14%, 6%, 12% and 4% for OA, BC, ammonium, sulfate and nitrate. The average density of a particle in the low FT was estimated to be 1.356 g cm$^{-3}$, following the method in Haslett et al. (2019a), assuming all particles are internally mixed. The density of $NH_4NO_3$ is assumed to be 1.725 g cm$^{-3}$ (Haslett et al., 2019a). When the additional $NH_4NO_3$ is internally mixed, it is estimated to lead to a ~4% increase in aerosol radius, assuming the particles are spherical. This is consistent with the slight vertical change of CMDs of bulk aerosols in the FT. It is likely that this internal mixing did not significantly alter the overall dry aerosol size distributions.

**Line 445** spay to spray

Accepted

**Supplement line 24** It might be nice to actually say what the background CO concentrations was in the FT.

FT CO background is calculated to be 78 ppbv (62 μg m$^{-3}$), which is summarized from the clean FT data (BC < 0.1 μg m$^{-3}$).

**Supplement lines 59–70** This analysis depends on either a lack of interference at m/z = 30 and m/z = 46 or on the frag table being correct. The former is shaky, since $CH_2O$ and $C_2H_6$ are at m/z = 30 and $CH_2O_2$ and $C_2H_6O$ are at m/z = 46. Given the highly oxidized aerosol, $CH_2O$ is likely to be plentiful. The frag table is of course designed for general ambient aerosol and can have striking problems with unusual composition. It is particularly troubling that the standard frag table for Squirrel 1.62G assumes there is nothing but $NO_2$ at m/z = 46. At m/z = 30 there is an indirect dependence on getting the m/z = 28 peak right. I haven't used a C-ToF, but that peak is so big on an HR-ToF that it seems to be a bit nonlinear. High CO from the aerosol can also screw that up, but I don't think your OA concentrations were that large.

We have added the discussion of uncertainty in the m/z30/46 ratio due to the possibility of organic interference in C-ToF AMS. This has been answered in the reply of "**Two related issues with nitrates: 1**".

**Reference**

Akagi, S. K., Craven, J. S., Taylor, J. W., McMeeking, G. R., Yokelson, R. J., Burling, I. R., Urbanski, S. P., Wold, C. E., Seinfeld, J. H., Coe, H., Alvarado, M. J., and Weise, D. R.: Evolution of trace gases and particles emitted by a chaparral fire in California, Atmos. Chem. Phys., 12, 1397–1421, https://doi.org/10.5194/acp-12-1397-2012, 2012.

[revised manuscript text omitted]

---

## Referee Report (RR1)

Review of: Vertical variability of the properties of highly aged biomass burning aerosol transported over the southeast Atlantic during CLARIFY-2017 Authors: Wu et al. Manuscript #: ACP-2020-197

The authors have provided a very comprehensive and thoughtful revision of their original manuscript. I appreciate the additional information that has been added. I only have a few minor comments. I was not an original reviewer for this manuscript, and after writing my review and revisiting the original comments, I came to a better understanding on some of the revisions. To me a few of the author revisions in response to the original reviewer comments did not always improve the writing. In places where my recommendations contradict those of the previous reviewers, I leave it to the authors and editor to arrive at a final decision. The line numbers below apply to the revised manuscript ("acp-2020-197-manuscript-version3.pdf").

Recommendation: Accept with Minor Revisions

Abstract, line 17-18: the comment "for the first time" doesn't seem quite right, as some members of this same team I believe documented a BBA plume at Ascension as part of SAFARI, and, the ORACLES measurements in 2016 preceeded the CLARIFY campaign. Some of that vertical structure is documented in Shinozuka et al., 2020 and Redemann et al. 2020. You could just leave out the phrase without loss of context, or, substitute something else here.

Page 1, line 48: Adebiyi and Zuidema 2016 might be the better reference here, as it focuses so strongly on the FT winds you mention.

Adebiyi, A., and Zuidema, P.: The Role of the Southern African Easterly Jet in Modifying the Southeast Atlantic Aerosol and Cloud Environments. Quarterly Journal of the Royal Meteorological Society, 142, 697, 1574–89. https://doi.org/10.1002/qj.2765, 2016

Page 2 line 56: I thought the point of Abel et al 2020 is that certain forms of mesoscale organization (POCs) don't support entrainment into the MBL so much. I'm not sure what to suggest here, but the sentence as written is mildly confusing.

Page 2 line 60: is there any evidence for the mechanism documented Fan et al., 2018, of ultra-fine particles enhancing shallow convection, at Ascension? We typically think of ultra-fine particle production over the marine ocean occurring over more pristine conditions (is my impression). The Koch and Del Genio reference is fine, but that is an overview document that highlights many other processes that are likely not relevant at Ascension. I would encourage the authors to include references here to processes that are more likely at Ascension. One mechanism I don't see mentioned is cloud dissipation in response to higher temperature/lower humidity. This effect was initially highlighted in Ackerman et al. 2000 and documented at Ascension in Zhang and Zuidema, 2019. The effect of enhancing convection formation through the additional heating is also used to explain a mid-morning cloudiness maximum at Ascension in Zhang and Zuidema, 2019. The free-tropospheric semi-direct effect also seems worth mentioning here, for which the authors could cite CLARIFY's very own Herbert et al., 2020 and Gordon et al., 2018 modeling papers. Under indirect effects, there are several pertinent papers that could be cited, e.g., Constantino and Breon, 2013; Kacarab et al., 2020; Gordon et al., 2018, so that is good, but nevertheless this section here would be improved by tying it in better to that discussion on p. 2.

Ackerman, A. S., Toon, O. B., Stevens, D. E., Heymsfield, A. J., Ramanathan, V., and Welton, E. J.: Reduction of tropical cloudiness by soot, Science, 288, 1042-1047, 2000.

Costantino, L. and Bréon, F.-M.: Aerosol indirect effect on warm clouds over South-East Atlantic, from co-located MODIS and CALIPSO observations, Atmospheric Chemistry and Physics, 13, 69–88, doi:10.5194/acp-13-69-2013, https://www.atmos-chem-phys.net/13/69/ 2013/, 2013.

Gordon, H., Field, P. R., Abel, S. J., Dalvi, M., Grosvenor, D. P., Hill, A. A., Johnson, B. T., Miltenberger, A. K., Yoshioka, M., and Carslaw, K. S.: Large simulated radiative effects of smoke in the south-east Atlantic, Atmospheric Chemistry and Physics, 18, 15 261–15 289, doi:10.5194/acp-18-15261-2018, https://www.atmos-chem-phys.net/18/15261/2018/, 2018

Gordon, H., Field, P. R., Abel, S. J., Barrett, P., Bower, K., Crawford, I., Cui, Z., Grosvenor, D. P., Hill, A. A., Taylor, J., Wilkinson, J., Wu, H., and Carslaw, K. S.: Improving aerosol activation in the double-moment Unified Model with CLARIFY measurements, Atmos. Chem. Phys. Discuss., https://doi.org/10.5194/acp-2020-68, accepted, 2020

Herbert, R. J., Bellouin, N., Highwood, E. J., and Hill, A. A.: Diurnal cycle of the semi-direct effect from a persistent absorbing aerosol layer over marine stratocumulus in large-eddy simulations, Atmos. Chem. Phys., 20, 1317–1340, https://doi.org/10.5194/acp-20-1317-2020, 2020

Kacarab, M., et al. "Biomass burning aerosol as a modulator of the droplet number in the southeast Atlantic region." *Atmospheric Chemistry and Physics*, vol. 20, no. 5, 2020, p. 3029

Mallet, M., Solmon, F., Nabat, P., Elguindi, N., Waquet, F., Bouniol, D., Sayer, A. M., Meyer, K., Roehrig, R., Michou, M., Zuidema, P., Flamant, C., Redemann, J., and Formenti, P.: Direct and semi-direct radiative forcing of biomass burning aerosols over the Southeast Atlantic (SEA) and its sensitivity to absorbing properties: a regional climate modeling study, Atmos. Chem. Phys. Discuss., https://doi.org/10.5194/acp-2020-317, in review, 2020

Zhang, J. and Zuidema, P.: The diurnal cycle of the smoky marine boundary layer observed during August in the remote southeast Atlantic. Atmos. Chem. Phys., 19, 14493-14516, doi:acp-19-14493-2019, 2019.

Input on previous reviewer's comments: Upon reading the previous reviewers' comments, I came to understand the Fan et al 2018 paper was recommended by a previous reviewer. It does not seem to me that the Fan et al mechanism has support from the CLARIFY measurements nor other modeling studies. I am not clear what the reviewer's recommendation was based on.

Page 2 line 65: the idea that the large spatial coverage of aerosol will generate a regional forcing of increasing importance with time isn't well supported in this sentence. One useful reference here might be

Kloster, D. Bachelet, M. Forrest, G. Lasslop, F. Li, S. Mangeon, J. R. Melton, C. Yue and J. T. Randerson N. Andela, D. C. Morton, L. Giglio, Y. Chen, G. R. van der Werf, P. S. Kasibhatla, R. S. DeFries, G. J. Collatz, S. Hantson, S.: A human-driven decline in global burned area. DOI: 10.1126/science.aal4108 Science 356 (6345), 1356-1362

Overall I would suggest the authors expend more time thinking critically on the previous reviewer comments, as opposed to simply accepting them.

p. 3 line 76: The Shinozuka et al 2020 paper would also be relevant to cite here, as it shows that there is too much BBA in the BL within most (all?) models, and, that the aerosol is located too low in altitude in most (all?) of the models (explaining the overestimate in the BL). This study also indicates a wide range of model SSA values, which is relevant to the subsequent sentence.

P. 3 line 84: the reason the Rajapakshe et al., 2017 paper reports an overestimate of the aerosol layer is primarily a reflection of the remote sensing algorithm, in which the aerosol layer base is overestimated (i.e. placed too high in altitude). I'm not sure I completely understand the point of this sentence or the previous sentence. Is it that we have little remote sensing information on the aerosol vertical structure from space that is known with confidence?

P. 3 line 90: I think the issue is that the lidar measurements are under constrained, and that the lidar perception of near- and far-field properties does not allow for a fully accurate extinction retrieval. It is not that they rely on assumptions on the aerosol properties. Maybe just say "…measurements due to retrieval limitations"? I can't think of a great reference here, as I don't believe a thorough analysis of the LASIC lidar data has been published, but Delgadillo et al 2018 do discuss the issues for this type of lidar.

Delgadillo, R., K. Voss and P. Zuidema, 2018: Characteristics of optically-thin coastal Florida cumuli from surface-based lidar measurements. *J. Geophys. Res.*, **123**, p. 10,591-10,605, doi:10.1029/2018JD028867

P. 4 line 114: "the BL" -> "a BL"

p. 4 line 119: how did Hywood et al. 2020 infer an aerosol age? Some more specificity to the aerosol age statements would be nice.

P.7 line 205: "data was" -> "data were"

P. 7 line 225: would suggest defining PM1 on line 207, where it Is made clear that PCASP measurements are based on diameters < 1 micron (as opposed to the PAS/CRDS).

p. 8 line 252: should "were" be put in the present tense to be consistent with the prior "is"?

P. 10, end: I would suggest moving the sentence that is currently on p. 11, line 329-331, to the end of this paragraph. Currently the paragraph feels disconnected and it is not clear to the reader until the next page what the implication of the combustion efficiency is for this study.

p. 13, line 394-396: I'm not quite following this sentence, can it be clarified? I think it is meant to communicate that LASIC SSA values are lower than CLARIFY BL SSA values.

p. 13, lines 397-406: Isn't the main finding here that CLARIFY FT SSA values agree with those from ORACLES, while the CLARIFY BL SSA values do not agree with those from LASIC, which are also measuring in the BL? what's not coming through in this paragraph, or the previous sentence, is that the SSA values within Zuidema 2018 were filter-based. Recall that the LASIC PSAP-derived absorption coefficients compare well with those from CLARIFY, and the difference in the LASIC-CLARIFY SSA measurements is in the extinction. So I don't think limitations with filter-based absorption explain the SSA differences, in contrast to what is written in the paragraph, with the Davies 2019 comparison only able to say something about the CLARIFY filter-derived values, not those from LASIC. Are the CLARIFY SSA values in the previous paragraph derived from the PAS/CRDS? I thought they were but either way might be good to restate here, to help the reader make more sense of the subsequent paragraph. If the CLARIFY SSA values are PAS/CRDS-derived, then the sentence focusing on Davies et al 2019 may not be that enlightening to this discussion. The LASIC and CLARIFY aerosol inlets have slightly different cut-offs (1.0 vs 1.3 aerodynamic diameter). Not clear if that explains it either. Perhaps the authors just want to provide an update on this comparison and say it is under investigation? Overall, please revisit this paragraph, to clarify further the distinction between the FT and BL comparisons.

P. 14 line 427: "could be also" -> "could also be" or "could be"

p. 14 line 431: "of well-mixed" -> "of a well-mixed"

p. 14 line 435: "C036, the" -> "C036, and the"

p. 14 line 459, 460: ""the increase in RH....an increase in aerosol scattering". grammatically, I think you need to either go with an "an" or a "the" in both places in front of "increase"

P. 16 line 491: "figure" - >"Figure"

p. 16 line 506: I don't think the comma is needed.

p. 17 lines 539-540: I don't understand how having an FT SSA that increases with altitude, and is lowest close to the cloud, enhances the direct radiative effect above what one expects from a column-mean SSA. At least, I think that is what the authors are implying.

P. 18 line 563: The new Mallet et al 2020 acpd manuscript would help to make this statement more concrete.

P. 18, line 578: Looks to me the sentence needs a "more" in front of the "likely" to be consistent with the "larger" at the beginning.

Figures:

Fig. 1: not a big deal but would be nice to see Ascension indicated in the lower 2 panels.

Fig. 2: would be good to spell out what conditions the 3 periods correspond to in the caption, for those readers who look at the figures first before reading the main text.

Fig. 10: would be good to include the date of the C036 flight in the caption, also for readers who look at figures prior to the text.

---

## Author Response (AR2)

**Response to reviewers**

**Firstly, we sincerely thank the editor and referees for their contributions to this work.**
**We have addressed the comments below. The original comments are in black, our replies are in blue and the changes in original manuscript are in red.**

**Anonymous Referee #3**

**General Comments:** The authors have provided a very comprehensive and thoughtful revision of their original manuscript. I appreciate the additional information that has been added. I only have a few minor comments. I was not an original reviewer for this manuscript, and after writing my review and revisiting the original comments, I came to a better understanding on some of the revisions. To me a few of the author revisions in response to the original reviewer comments did not always improve the writing. In places where my recommendations contradict those of the previous reviewers, I leave it to the authors and editor to arrive at a final decision. The line numbers below apply to the revised manuscript ("acp-2020-197-manuscript-version3. pdf").

Recommendation: Accept with Minor Revisions

**Detailed Comments:**
1. Abstract, line 17-18: the comment "for the first time" doesn't seem quite right, as some members of this same team I believe documented a BBA plume at Ascension as part of SAFARI, and, the ORACLES measurements in 2016 preceeded the CLARIFY campaign. Some of that vertical structure is documented in Shinozuka et al., 2020 and Redemann et al. 2020. You could just leave out the phrase without loss of context, or, substitute something else here.
We have removed "for the first time".

2. Page 1, line 48: Adebiyi and Zuidema 2016 might be the better reference here, as it focuses so strongly on the FT winds you mention. Adebiyi, A., and Zuidema, P.: The Role of the Southern African Easterly Jet in Modifying the Southeast Atlantic Aerosol and Cloud Environments. Quarterly Journal of the Royal Meteorological Society, 142, 697, 1574–89. https://doi.org/10.1002/qj.2765, 2016
We have added the reference of Adebiyi, A., and Zuidema, P. (2016).

3. Page 2 line 56: I thought the point of Abel et al 2020 is that certain forms of mesoscale organization (POCs) don't support entrainment into the MBL so much. I'm not sure what to suggest here, but the sentence as written is mildly confusing.
We have removed the reference of Abel et al, (2020) to avoid misleading the reader.

4. Page 2 line 60: is there any evidence for the mechanism documented Fan et al., 2018, of ultra-fine particles enhancing shallow convection, at Ascension? We typically think of ultra-fine particle production over the marine ocean occurring over more pristine conditions (is my impression). The Koch and Del Genio reference is fine, but that is an overview document that highlights many other processes that are likely not relevant at Ascension. I would encourage the authors to include references here to processes that are more likely at Ascension. One mechanism I don't see mentioned is cloud dissipation in response to higher temperature/lower humidity. This effect was initially highlighted in Ackerman et al. 2000 and documented at Ascension in Zhang and Zuidema, 2019. The effect of enhancing convection formation through the additional heating is also used to explain a mid-morning cloudiness maximum at Ascension in Zhang and Zuidema, 2019. The free-tropospheric semi-direct effect also seems worth mentioning here, for which the authors could cite CLARIFY's very own Herbert et al., 2020 and Gordon et al., 2018 modeling papers. Under indirect effects, there are several pertinent papers that could be cited, e.g., Constantino and Breon, 2013; Kacarab et al., 2020; Gordon et al., 2020. Further ahead I do see you discussing a few of these studies further, e.g., Gordon et al., 2018, so that is good, but nevertheless this section here would be improved by tying it in better to that discussion on p. 2.

We have extended the introduction of indirect and semi-direct effects, specifically for the southeast Atlantic studies.

These smoke layers typically over-lie vast stretches of marine stratocumulus clouds (Adebiyi et al., 2015), where they can exert a warming effect by absorbing both downwelling solar radiation and that scattered upwards from the low-lying clouds (Samset et al., 2013). This direct radiative effect is sensitive to the smoke's single-scattering albedo (SSA), which is a function of aerosol composition and size and evolves with particle age (Abel et al., 2005). Space-based and in-situ field observations also suggested that the smoke layers can be entrained into the marine boundary layer (MBL) during its transport from land over ocean (Painemal et al., 2014; Zuidema et al., 2018; Haslett et al., 2019a). *The entrained aerosols in the MBL can affect cloud microphysics by acting as cloud condensation nuclei (CCN), inducing indirect radiative effects over the southeast Atlantic by increasing cloud droplet number and reducing precipitation, thereby increasing cloud coverage and cloud albedo (e.g. Costantino and Bréon, 2013). In addition, BC immersed in cloud droplets absorbs light and may facilitate water evaporation. BC below clouds could enhance the formation of convection by providing additional heating within the sub-cloud layer. Zhang and Zuidema (2019) reported that shortwave absorption within the smoky MBL reduces the sub-cloud relative humidity due to raising the temperature, and so reduces daytime low-level cloud cover over the southeast Atlantic, which is opposite to the mechanism of increased aerosol increasing cloud droplet numbers. Furthermore, large eddy model studies have shown that marine stratocumulus clouds over the southeast Atlantic also adjust to the presence of overlying absorbing aerosol layers, depending on their properties and distance with low-cloud deck (e.g. Herbert et al., 2020). The above-cloud shortwave absorption can warm the FT, strengthening the temperature inversion and reducing the entrainment of warm and dry air from the FT into the MBL, thus influencing MBL humidity, temperature and dynamics. These effects described above, which perturb the temperature structure of the atmosphere and influence the cloud distribution, are collectively termed semi-direct effects.*

5. Page 2 line 65: the idea that the large spatial coverage of aerosol will generate a regional forcing of increasing importance with time isn't well supported in this sentence. One useful reference here might be

Kloster, D. Bachelet, M. Forrest, G. Lasslop, F. Li, S. Mangeon, J. R. Melton, C. Yue and J. T. Randerson N. Andela, D. C. Morton, L. Giglio, Y. Chen, G. R. van der Werf, P. S. Kasibhatla, R. S. DeFries, G. J. Collatz, S. Hantson, S.: A human-driven decline in global burned area. DOI: 10.1126/science.aal4108 Science 356 (6345), 1356-1362

Overall, I would suggest the authors expend more time thinking critically on the previous reviewer comments, as opposed to simply accepting them.

The referee has identified a lack of clarity in this sentence. We did not mean to discuss changes with time in this sentence, rather discuss the spatial extent of the aerosol influence. What we want to say is, the large spatial coverage of aerosol will generate an important regional forcing. We discussed the changes over time in the previous paragraph, "As controls continue to reduce aerosol emissions from fossil fuels and a changing climate potentially leads to more fires, the relative impact of BB on climate forcing is expected to increase (Fuzzi et al., 2015).", we also say "the contribution of transported BB aerosol to regional radiative forcing will also be increasingly important in the future".

We have removed "will be increasingly important" from the sentence highlighted by the reviewer to make sure the sentence is no longer misleading.

Although BB transport regions have lower aerosol concentrations than areas closer to the source, the large spatial coverage means that their contribution to the regional/global-average forcing is important.

6. P. 3 line 76: The Shinozuka et al 2020 paper would also be relevant to cite here, as it shows that there is too much BBA in the BL within most (all?) models, and, that the aerosol is located too low in altitude in most (all?) of the models (explaining the overestimate in the BL). This study also indicates a wide range of model SSA values, which is relevant to the subsequent sentence.

We have added this information.

A recent study demonstrated that models generally underestimate the smoke base height over the southeast Atlantic and thus lead to an overestimation of aerosol loading in the MBL (Shinozuka et al., 2019). Uncertainty in SSA is also one of the largest sources of uncertainty in estimating the aerosol direct effects (McComiskey et al., 2008; Shinozuka et al., 2019).

7. P. 3 line 84: the reason the Rajapakshe et al., 2017 paper reports an overestimate of the aerosol layer is primarily a reflection of the remote sensing algorithm, in which the aerosol layer base is overestimated (i.e. placed too high in altitude). I'm not sure I completely understand the point of this sentence or the previous sentence. Is it that we have little remote sensing information on the aerosol vertical structure from space that is known with confidence?

We agree with the reviewer and that is what we wanted to say - the aerosol layer base using the remote sensing is overestimated, and BB layer is placed too high in altitude. We have rephrased this.

Satellite-based observations have been employed in this region, but satellite retrievals often detect the bottom of the aerosol layer too high and thereby overestimate the above-cloud aerosol height (e.g. Rajapakshe et al., 2017). The ability of satellites to quantify BBA amount and its microphysical and optical properties in the marine BL is also limited, since the presence of intervening cloud layers brings significant challenges to retrievals of aerosol properties. Due to the persistent stratocumulus

cloud deck over the south Atlantic, most of the region is affected by clouds, and so MBL properties are hard to obtain from satellites.

8. P. 3 line 90: I think the issue is that the lidar measurements are under constrained, and that the lidar perception of near- and far-field properties does not allow for a fully accurate extinction retrieval. It is not that they rely on assumptions on the aerosol properties. Maybe just say "…measurements due to retrieval limitations" ? I can't think of a great reference here, as I don't believe a thorough analysis of the LASIC lidar data has been published, but Delgadillo et al 2018 do discuss the issues for this type of lidar.

We have rephrased this.

The vertically resolved retrievals obtained during the LASIC campaign using a co-located micropulse lidar also have retrieval limitations (Delgadillo et al., 2018).

9. P. 4 line 114: "the BL" -> "a BL"

We will keep "the BL" in manuscript, since we mean the specific BL over the southeast Atlantic.

10. p. 4 line 119: how did Haywood et al. 2020 infer an aerosol age? Some more specificity to the aerosol age statements would be nice.

We have added the information.

Haywood et al (2020) conducted back trajectories with particles released from Ascension Island at different altitudes from the MBL to FT, and they reported that air masses sampled in the CLARIFY operating area were of African BB origin and also indicated that the aerosol age was likely in a range of 4 to 10 days.

11. P.7 line 205: "data was" -> "data were" Accepted

12. P. 7 line 225: would suggest defining PM1 on line 207, where it Is made clear that PCASP measurements are based on diameters < 1 micron (as opposed to the PAS/CRDS).

Submicron aerosol (PM1) number concentrations from the PCASP were calculated using bins with diameter below 1 μm.

13. p. 8 line 252: should "were" be put in the present tense to be consistent with the prior "is"?

"the $NH_{4mea}^+$ is the measured ammonium concentration from the AMS, the $NH_{4neu}^+$ is the calculated ammunium concentration if all acids in the aerosol were neutralized" defines the terms $NH_{4neu}^+$ and $NH_{4mea}^+$, so we used present tense.

"The linear fitted $NH_{4mea}^+/NH_{4neu}^+$ ratios in the BL were $(0.86 \pm 0.01)$ and $(0.99 \pm 0.02)$ for period 1 and 3 respectively" presents measurement results from CLARIFY, and we consistently used past tense to describe observation results in the manuscript. So, we will keep the sentence as was written in the original manuscript.

14. P. 10, end: I would suggest moving the sentence that is currently on p. 11, line 329-331, to the end of this paragraph. Currently the paragraph feels disconnected and it is not clear to the reader until the next page what the implication of the combustion efficiency is for this study.

We have re-organized these two paragraphs.

The modified combustion efficiency (MCE) was calculated to indicate the combustion conditions at source (Yokelson et al., 2009). Details of the method of calculating MCE are listed in Supplementary S1. The MCEs of FT smoke were generally around 0.97 during CLARIFY, as shown in Fig. 5. An MCE > 0.9 is commonly used to indicate BB smoke predominantly influenced by combustion during the flaming phase whereas, MCE < 0.9 represents the smouldering phase (Reid et al., 2005). By this definition, CLARIFY smoke plumes transported from southern Africa are likely to be mostly controlled by flaming-phase combustion at source. The emission of BC is usually high during flaming combustion, while smouldering combustion tends to emit smoke high in CO and organic mass (e.g. Christian et al., 2003). The enhancement ratios of BC and OA with respect to CO (BC/$\Delta$CO and OA/$\Delta$CO, $\mu g\ m^{-3}$ / $\mu g\ m^{-3}$) are generally used to indicate the emission conditions of fire at source. For example, BC/$\Delta$CO values from 0.005 to 0.023 and OA/$\Delta$CO values from 0.037 to 0.066 were observed for BB source in flaming combustion from previous measurements (May et al., 2014; Pratt et al., 2011), while a lower range of (0.0014 – 0.0072) for BC/$\Delta$CO and a higher range of (0.080 – 0.096) for OA/$\Delta$CO, were reported for BB source in smouldering combustion (Capes et al., 2008; Kondo et al., 2011; May et al., 2014).

For CLARIFY, the BC/$\Delta$CO and OA/$\Delta$CO ratios ($\mu g\ m^{-3}$ / $\mu g\ m^{-3}$) were calculated in the FT by the unconstrained linear orthogonal distance regression (ODR) fit (Yokelson et al., 2013), and were calculated in the BL by dividing BC and OA by the excess concentration of CO, after background values had been removed (Lefer et al., 1994). The detailed calculation method is listed in Supplementary S1. The calculated enhancement ratios in FT and BL smoke for each flight are shown in Fig. 5.

15. p. 13, line 394-396: I'm not quite following this sentence, can it be clarified? I think it is meant to communicate that LASIC SSA values are lower than CLARIFY BL SSA values.

We have rephrased this sentence.

Ground-based in-situ SSA measurements made on Ascension Island in 2017 (Zuidema et al., 2018) are lower than CLARIFY BL SSA values, and are the lowest values compared to all previously reported observations of southern African BBA.

16. p. 13, lines 397-406: Isn't the main finding here that CLARIFY FT SSA values agree with those from ORACLES, while the CLARIFY BL SSA values do not agree with those from LASIC, which are also measuring in the BL? what's not coming through in this paragraph, or the previous sentence, is that the SSA values within Zuidema 2018 were filter-based. Recall that the LASIC PSAP-derived absorption coefficients compare well with those from CLARIFY, and the difference in the LASIC-CLARIFY SSA measurements is in the extinction. So I don't think limitations with filter-based absorption explain the SSA differences, in contrast to what is written in the paragraph, with the Davies 2019 comparison only able to say something about the CLARIFY filter-derived values, not those from LASIC. Are the CLARIFY SSA values in the previous paragraph derived from the PAS/CRDS? I thought they were but either way might be good to restate here, to help the reader make more sense of

the subsequent paragraph. If the CLARIFY SSA values are PAS/CRDS-derived, then the sentence focusing on Davies et al 2019 may not be that enlightening to this discussion. The LASIC and CLARIFY aerosol inlets have slightly different cut-offs (1.0 vs 1.3 aerodynamic diameter). Not clear if that explains it either. Perhaps the authors just want to provide an update on this comparison and say it is under investigation? Overall, please revisit this paragraph, to clarify further the distinction between the FT and BL comparisons.

We have rephrased this paragraph, emphasized the SSA consistence between CLARIFY and ORACLES FT values, and SSA difference between CLARIFY and LASIC BL values, and also the possible reason for the SSA difference between CLARIFY and LASIC.

These previous observations employed filter-based measurements, using the Particle Soot Absorption Photometer (PSAP) and nephelometer, in contrast to the PAS/CRD methods employed during CLARIFY. CLARIFY FT SSA values are similar to those measured from the FT during the ORACLES mission. It is also interesting to note that the radiometrically retrieved SSA from nine above-cloud flights performed during ORACLES in 2016 and 2017 (Cochrane et al., 2020; their Figure 4), which do not depend on in-situ measurements, yield average SSAs of (0.85 ± 0.02), (0.83 ± 0.03) and (0.82 ± 0.04) at wavelengths of 380, 550, and 660 nm respectively for FT BBA. These values are also in good agreement with our FT SSAs within the expected variability. However, CLARIFY BL SSA values do not agree with those from LASIC ground-based measurements. Although limitations with filter-based measurements of aerosol light absorption are known to introduce systematic measurement biases (Lack et al., 2008; Davies et al., 2019), the LASIC-derived aerosol absorption is comparable with those from the CLARIFY. The difference between CLARIFY and LASIC BL SSAs is possibly due to differences in the extinction measurements, which may be caused by the different inlet cut-offs (aerosol dynamic diameter of 1 μm for LASIC and 1.3 μm for CLARIFY).

17. P. 14 line 427: "could be also" -> "could also be" or "could be" Accepted

18. p. 14 line 431: "of well-mixed" -> "of a well-mixed" Accepted

19. p. 14 line 435: "C036, the" -> "C036, and the" Accepted

20. p. 14 line 459, 460: ""the increase in RH…..an increase in aerosol scattering". grammatically, I think you need to either go with an "an" or a "the" in both places in front of "increase" Accepted

21. P. 16 line 491: "figure" - >"Figure" Accepted

22. p. 16 line 506: I don't think the comma is needed. Accepted

23. p. 17 lines 539-540: I don't understand how having an FT SSA that increases with altitude, and is lowest close to the cloud, enhances the direct radiative effect above what one expects from a column-mean SSA. At least, I think that is what the authors are implying.

We have added more explanation here. As the SSA of aged BBA used in climate models is generally higher than the SSA in this study, the positive radiative direct effects above clouds may be underestimated. Since there is also a vertical variation in SSA, with much lower values at lower levels within the FT, this underestimation maybe more important for the cases with lower and thinner smoke layers above clouds. We also extend the discussion to include the implications for semi-direct effects.

The relatively low dry SSA measured during CLARIFY, as determined by highly sensitive and accurate measurements that are not subject to the artefacts of filter-based methods, is an important result. The SSA of aged BBA used in climate models is generally higher than the SSA in this study (e.g. Randles and Ramaswamy, 2010; Johnson et al., 2016; Herbert et al., 2020). Furthermore, the vertical profiles of SSA show that the lowest values (averages: 0.82, 0.81 and 0.79 at 405, 550 and 658 nm) occur at low FT layer around 2000 m altitude, immediately above the stratiform cloud. The air is also relatively dry within these low FT layers, meaning that the measured dry SSA is analogous to ambient condition. This is important as the positive radiative feedback associated with the aerosol direct effects may be underestimated in current models, especially for the cases with low and thin smoke layers above clouds. Herbert et al. (2020) also found that both the cloud response and semi-direct radiative effects increase for thinner and denser overlying aerosol layers with lower SSA. The bias in modelled SSA values is likely to lead to mis-representation of semi-direct effects as may neglecting the vertical variation in SSA. These findings suggest that modelled climate effects of BBA in this region need reassessment in future studies and the variation in SSA values in different BB regions should be considered.

24. P. 18 line 563: The new Mallet et al 2020 acpd manuscript would help to make this statement more concrete.
Considering these BBAs have a long lifetime and their spatial range spans thousands of kilometres, and the direct and semi-direct radiative effects of smoke layers in the southeast Atlantic area are highly sensitive to the absorbing properties of BBA (Mallet et al., 2020), modelled climate effects need re-assessment over this region.

25. P. 18, line 578: Looks to me the sentence needs a "more" in front of the "likely" to be consistent with the "larger" at the beginning. Accepted
Figures:
26. Fig. 1: not a big deal but would be nice to see Ascension indicated in the lower 2 panels. Accepted
27. Fig. 2: would be good to spell out what conditions the 3 periods correspond to in the caption, for those readers who look at the figures first before reading the main text. Accepted
28. Fig. 10: would be good to include the date of the C036 flight in the caption, also for readers who look at figures prior to the text. Accepted